# Text Promptable Surgical Instrument Segmentation with Vision-Language Models

**Zijian Zhou**[1]    **Oluwatosin Alabi**[2]    **Meng Wei**[2]    **Tom Vercauteren**[2]    **Miaojing Shi**[3✉]

[1]Department of Informatics, King's College London
[2]School of Biomedical Engineering & Imaging Sciences, King's College London
[3]College of Electronic and Information Engineering, Tongji University
{first_name}.{last_name}@kcl.ac.uk; mshi@tongji.edu.cn

## Abstract

In this paper, we propose a novel text promptable surgical instrument segmentation approach to overcome challenges associated with diversity and differentiation of surgical instruments in minimally invasive surgeries. We redefine the task as text promptable, thereby enabling a more nuanced comprehension of surgical instruments and adaptability to new instrument types. Inspired by recent advancements in vision-language models, we leverage pretrained image and text encoders as our model backbone and design a text promptable mask decoder consisting of attention- and convolution-based prompting schemes for surgical instrument segmentation prediction. Our model leverages multiple text prompts for each surgical instrument through a new mixture of prompts mechanism, resulting in enhanced segmentation performance. Additionally, we introduce a hard instrument area reinforcement module to improve image feature comprehension and segmentation precision. Extensive experiments on several surgical instrument segmentation datasets demonstrate our model's superior performance and promising generalization capability. To our knowledge, this is the first implementation of a promptable approach to surgical instrument segmentation, offering significant potential for practical application in the field of robotic-assisted surgery. Code is available at https://github.com/franciszzj/TP-SIS.

## 1   Introduction

Minimally invasive surgeries (MIS) have gained widespread attention in various surgical disciplines [16, 48] due to their benefits over traditional open surgery, such as reduced patient discomfort and faster recovery times. Nonetheless, the restricted field of view and indirect vision through endoscopic cameras for existing MIS procedures presents considerable obstacles, making the development of robot-assisted MIS increasingly crucial. Current robot-assisted surgery has to operate under direct control of the surgeon. Enhancing the automatic comprehension of the surgical process through precise instrument segmentation is seen as an essential building block to introduce automation and facilitate robot-surgeon interaction. Despite this recognised importance, existing automatic surgical instrument segmentation faces significant challenges.

First, with fast-paced advances in MIS, there is a surge in the variety of surgical instruments from different vendors. This is however compounded with the lack of a comprehensive and large-scale dataset dedicated to the learning of surgical instrument segmentation. Current methods [43, 20, 12, 55, 5] do not adequately adapt to the ever-changing set of surgical instruments, necessitating re-labelling and re-training of models with the introduction of each new instrument. This has notably hampered the practical application of surgical instrument segmentation within the MIS field.

---

✉ Corresponding author.

37th Conference on Neural Information Processing Systems (NeurIPS 2023).

Second, the segmentation methods face difficulty in distinguishing between different categories of instruments [5], which often have similar appearances. The subtle visual difference across instruments and the often difficult imaging conditions in the confines of the surgical cavity make it harder to differentiate between instrument categories, causing poor segmentation performance.

Overcoming these challenges requires learning-based approaches that are more flexible and robust than the current state of the art. Recent progress in pre-trained vision-language models [33, 36, 47] offer novel opportunities for our research questions as demonstrated by results obtained in diverse downstream computer vision tasks [10, 35, 44, 30, 29, 46, 9]. On the one hand, these vision language models, trained on abundant data, offer robust features which can help compensate for the scarcity of surgical instrument segmentation data. On the other hand, the capability of these models to output aligned image and text features allows for text to supplement information for surgical instrument segmentation, thus simplifying the differentiation between various instrument types.

To address the first challenge, we observe that existing supervised instrument segmentation models [12, 55, 5, 3] are trained on predefined categories, thereby limiting their semantic understanding of instruments outside of the narrow training set. Drawing inspiration from recent advancements in text promptable image segmentation [29, 46] with vision-language models [36], we redefine the task as a text promptable surgical instrument segmentation, thereby enhancing generalization and adaptability to an ever-growing array of new surgical instruments. Figure 1 contrasts the predefined category method with our text-promptable approach. Our model adapts the image and text encoders from a pretrained vision-launguage model, CLIP [36], to extract features from both the surgical image and textural prompt; *text promptable mask decoder* consisting of attention-based and convolution-based prompting schemes is specifically designed to prompt the surgical instrument in the image from coarse to fine.

Varying text descriptions for each surgical instrument can yield distinct segmentation results. To tackle it, we propose a *mixture of prompts mechanism*, inspired by mixture of experts (MoE) [31, 41, 37], to leverage diverse prompts. This mechanism inputs multiple source prompts to the model and fuses the resulting predictions via weights generated from a visual-textural gating network. The final segmentation result hence benefits from various prompt information.

To address the second challenge, we aim to enhance the model's capacity to obtain improved image features, thereby reinforcing its ability to segment various surgical instrument categories and delineate precise edge details. We propose a *hard instrument area reinforcement module* intertwined with the popular image reconstruction approach, masked autoencoder (MAE) [15]. This module performs hard area mining at the site of segmentation error; by masking out the hard area, the module injects an auxiliary image reconstruction to the segmentation model to enahce the understanding of difficult areas in the image features.

To the best of our knowledge, we introduce the first text promptable surgical instrument segmentation approach. Our extensive experiments demonstrate state-of-the-art performance on several surgical instrument segmentation datasets [1, 2, 39, 17]. Additionally, we evaluate the generalization capability of our model through cross-validation between the two datasets, showcasing promising results and emphasizing the significant application potential of our method.

## 2  Related Work

**Surgical instrument segmentation.** Most of the current works on surgical instrument segmentation are traditional vision-based models. For instance, TernausNet [43] trains an optimized U-Net [38] model for the segmentation of a restricted variety of surgical instruments. ISINet [12] proposes an instance-based segmentation method that utilizes a temporal consistency module to identify instrument candidates. MF-TapNet [20] utilizes motion flow information through an attention pyramid network and is trained independently for binary, part, and instrument type segmentation tasks. While most approaches rely on convolutional networks, a few recent works [55, 3] have investigated the use of transformer-based methods. For example, MATIS [3] utilizes pixel-wise attention, masked attention modules targeting for instrument areas, and video transformers for temporal information handling. Although the segmentation accuracy has been significantly improved by incorporating various architectures and modules, existing works are limited by their reliance on categorical understanding from the annotated training data; when new categories are introduced, they will have to be re-trained, which requires demanding labelling efforts and computational resources.

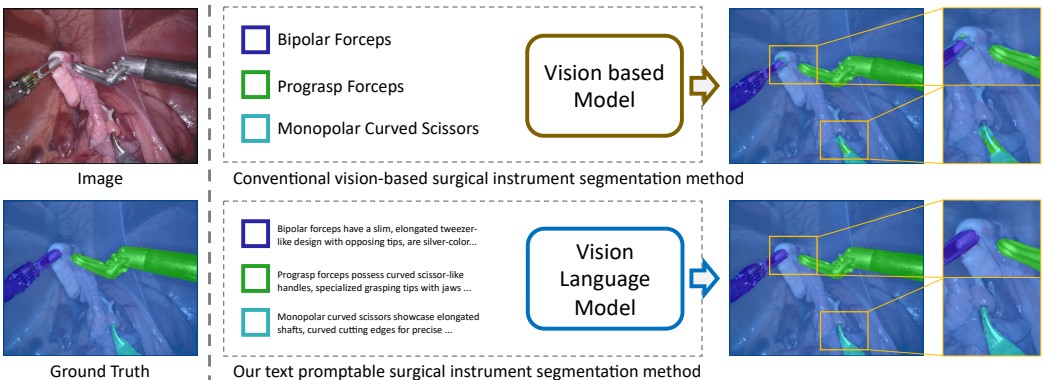

Figure 1: Left: input image and its ground truth. Top Right: conventional vision-based surgical instrument segmentation model with predefined categories. Bottom Right: ours leveraging vision-language model for text promptable surgical instrument segmentation.

**Vision-language models.** Large pretrained models like CLIP [36] has emerged as the most promising approaches for achieving state of the art results on downstream computer vision tasks. CLIP [36] is a large pretrained vision-language model [45] that aligns text with image modalities in the embedding space to solve downstream tasks like zero-shot image classification [36, 27]. It has been successfully applied to various vision tasks, including video captioning [44], video-text retrieval [30, 10], image generation [35], relation learning [57] and text promptable segmentation [46, 29].

**Text promptable segmentation.** Text promptable segmentation, also conceptualized as referring expression segmentation in the literature, involves the utilization of natural language expressions as prompts for image segmentation, deviating from the traditional reliance solely on class label annotations for training images [18]. Early works use convolution neural networks (CNNs) and recurrent neural networks (RNNs) to extract visual and textual features which are fused via concatenation and used to perform segmentation [18, 25]. Attention mechanisms are also introduced for exploiting relations between visual and textual features [18, 25, 42, 52]. More recent approaches utilize transformers to perform visual and textual feature fusion either in the encoder [11, 49, 51, 22, 54] or decoder [46, 29, 7].

Large pretrained vision-language models have also been utilized for this task [46, 29, 53, 28, 24]. For instance, CRIS [46] uses the CLIP image and text encoders for multimodal knowledge transfer and a transformer decoder for multimodal fusion and segmentation. CLIPSeg [29] extends prompt-based segmentation beyond text prompts to include image prompts processed by the CLIP image encoder. A recent concurrent work, SAM [24], introduces a transformer-based model capable of generating object masks using either image or text prompts. However, existing promptable segmentation methods are not designed to solve highly specific tasks like surgical instrument segmentation.

## 3 Method

In this section, we begin by defining the task (Section 3.1), then introduce the image and text encoders inherited from a pretrained vision-language model (CLIP [36]), serving as the feature extractors of our method (Section 3.2). Next, we develop a text promptable mask decoder responsible for decoding the visual features into instrument segments with the help of textual features (Section 3.3). Finally, we elaborate on two key modules: the mixture of prompts module (Section 3.4), designed to enhance segmentation performance through multiple prompts; and the hard instrument area reinforcement module (Section 3.5), focused on improving instrument segmentation on difficult parts.

### 3.1 Task Definition

Given an image $I \in \mathbb{R}^{H \times W \times 3}$ consisting of surgical instruments for an endoscopic operation, as well as a text description $T$ describing the name, appearance and function of a given instrument, the objective is to obtain the surgical instrument mask $M \in \{0, 1\}^{H \times W}$ from $I$, prompted by the instrument description $T$. In cases when multiple instruments need to be segmented, we input the model with

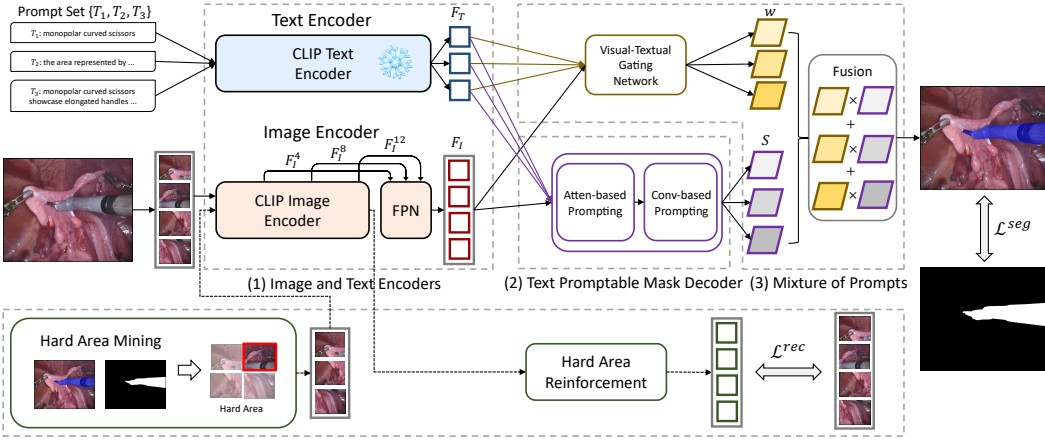

Figure 2: An overview of our method. Our method comprises four key modules: 1) image and text encoders derived from the pretrained vision-language model to obtain visual and textual features; 2) a text promptable mask decoder consisting of attention- and convolution-based prompting schemes for predicting the score map from image features through text prompts; 3) a mixture of prompts mechanism that utilizes a visual-textual gating network to produce pixel-wise weights for merging different score maps; 4) a hard instrument area reinforcement module to reinforce image representation learning specifically on hard-predicted area.

multiple text descriptions corresponding to different categories, and the goal becomes producing the segmentation result $M \in \{0, ..., C\}^{H \times W}$, where $C$ represents the number of instrument categories.

## 3.2 Image and Text Encoders

To accomplish text promptable surgical instrument segmentation, we first encode the input endoscopic image $I$ and instrument description $T$, obtaining the visual feature $F_I$ and textual feature $F_T$, respectively. Importantly, to enforce the semantic congruity between $F_I$ and $F_T$, we leverage the strength of the image and text encoders from the popular vision-language model, CLIP [36], pretrained on natural images.

**Image encoder.** Given the input image $I$, we utilize and fine tune the image encoder from CLIP [36], which is a vision transformer (ViT) [8], for feature extraction. The ViT-based image encoder comprises 12 layers. We use $F_I^l \in \mathbb{R}^{N \times D}$ to denote its $l$-th layer output, where $l$ ranges from 1 to 12, $N$ is the number of visual tokens, and $D$ is the dimension of visual features in ViT. Supposing the patch size of the input image is $p$, then $N = \frac{H}{p} \times \frac{W}{p}$.

*Multi-scale feature augmentation.* Extracting multi-scale features is important for semantic segmentation in order to segment objects in different scales. Whilst the image encoder in CLIP is not originally designed for multiple scales, we equip it with this ability. Specifically, simulating the multi-scale extraction from CNN-based architecture, we take the output features from the 4-th, 8-th, and 12-th layers of the ViT, *i.e.* $F_I^4$, $F_I^8$, $F_I^{12}$, representing a wide coverage of level-of-detail for the image; and fuse these features by a feature pyramid network (FPN) [26]. Notice in order to feed these features into FPN, we need to first reshape them into the size of $\frac{H}{p} \times \frac{W}{p} \times D$, then we up-sample $F_I^4$ and down-sample $F_I^{12}$ both by a factor of 2. Finally, the output of the FPN is reshaped to size of $N \times D$ to obtain the multi-scale image feature $F_I \in \mathbb{R}^{N \times D}$.

**Text encoder.** Given the input description $T$, we utilize and keep frozen the text encoder from CLIP, which is a transformer encoder [45], for feature extraction. Within the text encoder, $T$ is tokenized with the inclusion of a [CLS] token at its beginning, representing the global information of the description. Following the common practice in [46, 29], we use the corresponding feature from the [CLS] token as the global textual feature for the description. We denote it by $F_T \in \mathbb{R}^{1 \times D}$, where $D$ is dimension of the textual feature, equivalent to that of the visual feature.

### 3.3 Text Promptable Mask Decoder

After obtaining $F_I$ and $F_T$, our goal is to decode from $F_I$ a score map $S \in \mathbb{R}^{H \times W}$ with the help of $F_T$. Each pixel of $S$ signifies the probability of this pixel belonging to the instrument class described in $F_T$. By thresholding $S$, it can be transformed into the desired mask $M$. To achieve this goal, we present two prompting schemes: attention-based prompting and convolution-based prompting, specified below. For a detailed structure figure of the text promptable mask decoder, please refer to the supplementary material.

**Attention-based prompting.** Inspired by [45], we employ the attention mechanisms to attend the instrument area in $F_I$ using $F_T$. First, we compute the self-attention (SA) [45] within $F_I$ to promote the foreground area in the endoscopic image. Layer normalization (LN) [4] is applied to $F_I$ beforehand while skip connection [14] is applied afterwards. We write out this process: $F_{I_S} = \text{SA}(\text{LN}(F_I)) + F_I$. Next, we compute the cross-attention (CA) [45] between $F_{I_S}$ and $F_T$ to localize the instrument area in $F_{I_S}$ according to $F_T$. Similarly, this process is written as: $F_{I_C} = \text{CA}(\text{LN}(F_{I_S}), F_T) + F_{I_S}$. Subsequently, we apply the feed-forward network, which is basically a fully connected layer to further polishing $F_{I_C}$, $F_{I_F} = \text{FFN}(\text{LN}(F_{I_C})) + F_{I_C}$. SA, CA and FFN are organized in one decoding block, we devise three such blocks sequentially to obtain the attention-based prompted feature $F_{I_A} \in \mathbb{R}^{N \times D}$.

**Convolution-based prompting.** We design a novel convolution-based prompting scheme to further refine $F_{I_A}$ by using $F_T$ to convolve it locally. To do this, $F_{I_A} \in \mathbb{R}^{N \times D}$ needs to be firstly reshaped to restore its spatial dimensions, *i.e.* $\tilde{F}_{I_A} \in \mathbb{R}^{\frac{H}{p} \times \frac{W}{p} \times D}$. Our idea is to transform $F_T$ into the convolution parameters, *i.e.* kernel weights and bias, and use these parameters to convolve $\tilde{F}_{I_A}$, such that the instrument-related positions will be promoted. The transformation is done via a fully connected (FC) layer: $\tilde{F}_T = FC(F_T)$. $F_T$ is a vector of dimension $D$, the FC layer reshapes it to the vector $\tilde{F}_T$ of dimension $D \times k \times k + 1$, with "$k \times k$" representing the convolution kernel size and "$+1$" is the extra dimension accounting for bias. This allows decomposition of $\tilde{F}_T$ into convolution weights $w \in \mathbb{R}^{1 \times D \times k \times k}$ and bias $b \in \mathbb{R}^1$, which are subsequently used to convolve $\tilde{F}_{I_A}$,

$$S = \text{Sigmoid}(\text{Conv}(\tilde{F}_{I_A}|w, b)). \tag{1}$$

Sigmoid is applied to transform the final output into probabilities, hence producing the score map $S$.

The attention-based prompting computes the attention at each pixel by interacting it with pixels from the whole feature map, while in the convolution-based prompting, each pixel only interacts with its neighboring pixels in the $k \times k$ convlution kernel. We can see the former prompting as a global decoding process while the latter as a local decoding process to refine the former.

### 3.4 Mixture of Prompts

The provision of the prompt is crucial for the surgical instrument segmentation accuracy. Recognizing the impact of different prompts on a given model [56, 50], we propose to effectively leverage the strength of various prompts to facilitate the recognition of surgical instruments. Inspired by mixture of experts (MoE) [21], which divides a complex task into multiple sub-tasks and distributes them amongst multiple experts, we introduce a mixture of prompts (MoP) scheme to combine the segmentation outputs from multiple prompts via a visual-textual gating network.

**Acquisition of prompts.** There exist multiple strategies for generating prompts from simple to more advanced ones. We developed three types of prompt with increasing complexity. First, we use the name of each instrument class as the prompt. Second, inspired by CLIP's prompt template, we design a similar prompt template in the surgical context, *i.e.* "the surgical instrument area represented by the [class name]". Third, we leverage the recent released large language model GPT-4 [34] to generate prompts. We pose a question template: "Please describe the appearance of [class name] in endoscopic surgery, change the description to a phrase with a subject, and do not use colons." Please see supplementary material for the full list of our prompts.

**Fusion of prompt predictions.** Given the prompt set $\mathcal{T} = \{T_1, T_2, T_3\}$ for an instrument to be segmented, we denote the corresponding textual feature for $i$-th prompt as $F_T^i$. Each $F_T^i$ is paired the visual feature $F_I$ to predict a score map $S_i$ as outlined in Section 3.3, resulting in a set of score maps $\mathcal{S} = \{S_1, S_2, S_3\}$ for all prompts. We have parallelled the prompt processing by reshaping different

prompts to the batch dimension, exploiting the transformer structure to concurrently obtain $\mathcal{S}$. To combine the score maps in $\mathcal{S}$, we devise a visual-textual gating network to predict the pixel-wise weight maps.

*Visual-textual gating network $\mathcal{G}$.* $\mathcal{G}$ consists of a 3-layer residual block [14]. It ingests both the visual feature $F_I$ and textual feature $F_T^p$: we duplicate $F_T \in \mathbb{R}^{1 \times D}$ to $F_T^p \in \mathbb{R}^{N \times D}$ to match $F_I$'s dimension and concatenate them before inputting to $\mathcal{G}$. The outputs of $\mathcal{G}$ are three weight maps corresponding to the three score maps in $\mathcal{S}$. These weights are normalized using softmax operations along the prompt dimension. We calculate the weighted sum of scores maps in $\mathcal{S}$ to derive the final score map $S^{\text{et}} \in \mathbb{R}^{H \times W}$. The final mask prediction $\mathcal{M}$ is obtained from $S^{\text{et}}$ by thresholding it.

### 3.5 Hard Instrument Area Reinforcement

Current model falls short on distinguishing the accurate category and boundary of surgical instruments due to the complex surgical conditions (*e.g.* lighting variance) in endoscopy scenarios. We seek to reinforce its performance on the hard-predicted area in an image by utilizing a MAE-like structure [15] to reconstruct the image especially on hard-predicted area for representation enhancement. We propose a hard instrument area reinforcement module to bolster the model's segmentation accuracy for various instrument classes and their intricate details. Unlike MAE, which pays equal attention to all areas, our objective is to focus the visual encoder on discriminating the challenging instrument area. Therefore, we first introduce a hard area mining scheme to construct a masked image, then feed the visible patches into the decoder to reconstruct the whole image. This module shares the same image encoder with segmentation model.

**Hard area mining.** Given the estimated segmentation mask $M^{\text{et}}$, we compare it with the ground truth mask $M^{\text{gt}}$ to obtain the falsely predicted area, which is taken as the hard instrument area for segmentation. For the input image $I$, we divide the area of hard regions over that of the whole image to obtain a masking ratio $r \in (0, 1)$: 1) if $r$ exceeds a pre-defined threshold $r_t$, we randomly unmask some masked pixels till the ratio falls to $r_t$ to create the masked image; 2) if $r$ is less than $r_t$, we randomly mask some more unmasked pixels till the ratio meets $r_t$ to create the masked image.

**Hart area reinforcement.** We feed the obtained masked image into the image encoder and employ the same decoder structure as MAE to reconstruct the full image area based on the unmasked area. This helps the model to focus on recognizing the subtle details of the hard-predicted area.

It is worth mentioning that this module is no longer needed during testing.

### 3.6 Model Training

Our method is based on the pre-trained vision-language model, CLIP. During training, given limited surgical instrument descriptions, we freeze the text encoder and only finetune the image encoder. Two distinct loss functions, $\mathcal{L}^{seg}$ for image segmentation and $\mathcal{L}^{rec}$ for hard instrument area reinforcement, are introduced.

For $\mathcal{L}^{seg}$, binary cross entropy loss is applied:

$$\mathcal{L}^{seg} = -\sum_j (M_j^{\text{gt}} \log M_j^{\text{et}} + (1 - M_j^{\text{gt}}) \log(1 - M_j^{\text{et}})) \tag{2}$$

where $M_j^{\text{gt}}$ and $M_j^{\text{et}}$ denote the labels of ground truth mask $M^{\text{gt}}$ and prediction mask $M^{\text{et}}$ at the $j$-th position, respectively.

For $\mathcal{L}^{rec}$, denoting by the $I^{\text{rec}}$ and $I$ the reconstructed image and original image, we employ the L2 loss to minimize their pixel-wise distance:

$$\mathcal{L}^{\text{rec}} = \sum_j \| I_j^{\text{rec}} - I_j \|^2 \tag{3}$$

where $I_j^{\text{rec}}$ and $I_j$ represent the reconstructed and original pixel values at $j$-th location, respectively.

The overall loss function is given by $\mathcal{L} = \mathcal{L}^{seg} + \lambda \mathcal{L}^{rec}$, $\lambda$ serves as the loss weight.

Table 1: Comparison between our method and other state-of-the-art methods on the EndoVis2017 dataset. Methods in the first (top) group are conventional supervised methods with predefined categories; methods in the second group (bottom) are text promptable using vision-language models. Cross-Ours in the last row represents the cross-dataset experiment.

| Method | Ch_IoU | ISI_IoU | BF | PF | LND | VS | GR | MCS | UP | mc_IoU |
|---|---|---|---|---|---|---|---|---|---|---|
| TernausNet-11 [43] | 35.27 | 12.67 | 13.45 | 12.39 | 20.51 | 5.97 | 1.08 | 1.00 | 16.76 | 10.17 |
| MF-TAPNet [20] | 37.35 | 13.49 | 16.39 | 14.11 | 19.01 | 8.11 | 0.31 | 4.09 | 13.40 | 10.77 |
| ISINet [12] | 55.62 | 52.20 | 38.70 | 38.50 | 50.09 | 27.43 | 2.01 | 28.72 | 12.56 | 28.96 |
| TraSeTR [55] | 60.40 | 65.20 | 45.20 | 56.70 | 55.80 | 38.90 | 11.40 | 31.3 | 18.20 | 36.79 |
| S3Net [5] | 72.54 | 71.99 | **75.08** | 54.32 | 61.84 | 35.5 | **27.47** | 43.23 | **28.38** | 46.55 |
| MATIS [3] | 71.36 | 66.28 | 68.37 | 53.26 | 53.55 | 31.89 | 27.34 | 21.34 | 26.53 | 41.09 |
| CRIS [46] | 69.94 | 67.83 | 54.87 | 50.21 | 68.33 | 50.12 | 0.00 | 43.97 | 0.00 | 38.21 |
| CLIPSeg [29] | 70.15 | 65.02 | 51.29 | 42.27 | 49.56 | 30.12 | 9.96 | 30.69 | 20.05 | 33.42 |
| **Ours (448)** | 77.79 | 76.45 | 69.57 | 68.91 | 89.88 | 82.60 | 0.00 | 72.53 | 0.00 | 54.78 |
| **Ours (896)** | **79.90** | **77.83** | 68.58 | **73.52** | **92.74** | **83.90** | 0.13 | **74.70** | 0.00 | **56.22** |
| **Cross-Ours (896)** | 72.18 | 70.44 | 65.54 | 58.19 | 84.01 | 67.30 | 0.06 | 68.47 | 0.06 | 49.09 |

Table 2: Comparison between our method and other state-of-the-art methods on the EndoVis2018 dataset.

| Method | Ch_IoU | ISI_IoU | BF | PF | LND | SI | CA | MCS | UP | mc_IoU |
|---|---|---|---|---|---|---|---|---|---|---|
| TernausNet-11 [43] | 46.22 | 39.87 | 44.20 | 4.67 | 0.00 | 0.00 | 0.00 | 50.44 | 0.00 | 14.19 |
| MF-TAPNet [20] | 67.87 | 39.14 | 69.23 | 6.10 | 11.68 | 14.00 | 0.91 | 70.24 | 0.57 | 24.68 |
| ISINet [12] | 73.03 | 70.97 | 73.83 | 48.61 | 30.98 | 37.68 | 0.00 | 88.16 | 2.16 | 40.21 |
| TraSeTR [55] | 76.20 | - | 76.30 | 53.30 | 46.50 | 40.60 | 13.90 | 86.30 | 17.50 | 47.77 |
| S3Net [5] | 75.81 | 74.02 | 77.22 | 50.87 | 19.83 | 50.59 | 0.00 | 92.12 | 7.44 | 42.58 |
| MATIS [3] | 84.26 | 79.12 | 83.52 | 41.90 | 66.18 | 70.57 | 0.00 | **92.96** | 23.13 | 54.04 |
| CRIS [46] | 74.10 | 72.29 | 73.58 | 58.20 | 47.64 | 72.14 | 4.56 | 45.99 | 20.18 | 46.04 |
| CLIPSeg [29] | 74.95 | 69.86 | 67.25 | 39.59 | 36.72 | 47.27 | 2.92 | 79.96 | 4.22 | 39.7 |
| **Ours (448)** | 82.67 | 81.54 | 81.53 | 70.18 | 71.54 | 90.58 | 21.46 | 65.57 | **57.51** | **65.48** |
| **Ours (896)** | **84.92** | **83.61** | **84.28** | 73.18 | 78.88 | 92.20 | 23.73 | 66.67 | 39.12 | 65.44 |
| **Cross-Ours (896)** | 66.25 | 64.92 | 65.81 | 56.12 | 44.72 | 79.77 | 1.22 | 8.97 | 4.77 | 37.34 |

# 4 Experiments

## 4.1 Datasets and Metrics

**Datasets** We evaluate our method on two endoscopic surgical instrument segmentation datasets: EndoVis2017 [1], EndoVis2018 [2]. EndoVis2017 includes 10 videos from the da Vinci robotic system, containing 6 distinct surgical instrument types (bipolar forceps, prograsp forceps, large needle driver, vessel sealer, grasping retractor, monopolar curved scissors) and an ultrasonic probe (classified as other instruments). Following [43], we employ 4-fold cross-validation to assess the model performance. For EndoVis2018, we adopt the widely-used labeling and dataset partitioning method proposed in [12]. This dataset consists of 15 video sequences, with 11 training and 4 testing sequences, and 7 predefined instrument categories (bipolar forceps, prograsp forceps, large needle driver, monopolar curved scissors, ultrasound probe, suction instrument, clip applier). Notably, two categories (suction instrument and clip applier) differ from those in EndoVis2017 (vessel sealer and grasping retractor), enabling cross-category experiments to evaluate our model's generalizability. Additionally, both datasets provide binary and parts segmentation labels. Binary segmentation comprises background tissue and instruments, while parts segmentation distinguishes instrument components as shaft, wrist, and claspers. Besides the two datasets, we have also evaluated our method on the EndoVis2019 [39] and Cholecseg8k [17] datasets in the supplementary material.

**Metrics** Adopting the evaluation method from [12], we utilize three prevalent IoU-based evaluation metrics: Ch_IoU, ISI_IoU, and mc_IoU. 1) Ch_IoU computes the mean IoU for each category present in the ground truth of an image, then averages them across all image. 2) ISI_IoU extends Ch_IoU by computing mean IoUs for all predicted categorizes regardless of their presence in the ground truth of the image. Ch_IoU is normally greater than or equal to ISI_IoU. 3) mc_IoU is a measurement addressing category imbalance by changing the averaging order in ISI_IoU.

## 4.2 Implement Details

**Vision-Language model.** We employ the image and text encoders from the widely-used CLIP model [19], specifically the ViT-B-16 variant pre-trained on the Laion2B [40] dataset, to construct

our promptable surgical instrument segmentation model. The image encoder has a patch size of 16 ($p = 16$) and the text encoder has a token length limit of 77.

**Hyper-parameters.** We offer two training/inference default image sizes, $896 \times 896$ and $448 \times 448$, which are compatible with the size requirements for image patching in the ViT-based image encoder [8]. During evaluation, we restore the segmentation prediction to the original image size. Feature dimension $D$ is 1024. Following [46], we employ a threshold $\theta = 0.35$ to transform the score map $S$ into mask $M$, and select the highest-scoring category per pixel in multi-category cases. For hard instrument area reinforcement, we set mask ratio $r$ to 0.25. Finally, the loss weight $\lambda$ is 0.5. All hyperparameters are determined empirically by segregating 20% of the training data as a validation set following [46].

**Training.** In our experiments, we adopt the Adam [23] optimizer with a learning rate of $1e^{-4}$. We train for 50 epochs, reducing the learning rate to $1e^{-5}$ at the 35-th epoch. To enhance the model's generalization, we apply data augmentation techniques to the image, including random crop, horizontal flip, random rotation, and brightness perturbation. The model is trained on 4 V100 GPUs, the batch size is 16.

### 4.3 Results

**Comparison to state of the art.** In Table 1 and Table 2, we compare our method with a series of state of the art methods on the EndoVis2017 and EndoVis2018 datasets, respectively. The methods can be divided into two group depending on whether they have utilized textural prompts for surgical instruments. Ours belongs to the second group. As demonstrated by the results, our method significantly surpasses the state of the art in the first group: *e.g.* on the EndoVis2017 dataset, +7.36% on Ch_IoU, +5.84% in ISI_IoU, and +9.67% in mc_IoU. Particularly, a smaller difference between Ch_IoU and ISI_IoU for our method indicates it has fewer misclassified categories compared to other methods. Next, comparing to methods in second group, which are originally designed for natural image segmentation, our method yields clearly superior results, validating the effectiveness of our specifically tailored modules for surgical instrument segmentation. Notably, we implement both CRIS and CLIPSeg using their open-source implementations with batch size 64 for training.

*Comparison to SAM.* Whilst we were working on our work, the segment anything model (SAM) [24] was released concurrently. SAM possesses the capability to output corresponding object masks through visual or text prompts. Its officially released implementation however only contains visual prompts (point, box and mask), lacks text prompting. We thus leverage a community-based solution, lang-segment-anything [32], which enables this function for SAM. We found that this SAM variant struggles with medical prompts (*e.g.* the Ch_IoU is only 17.77% and 22.08% on EndoVis2018 and EndoVis2017, respectively), performing significantly inferior than ours. For more details, we refer readers to the supplementary material.

**Cross-dataset experiment.** We also evaluate our method in the cross-dataset setting by training it on EndoVis2018 and testing it on EndoVis2017, and vice versa. As depicted in Tables 1 and Table 2 (*i.e.* Cross-Ours), our method still gets competitive results compared with the state of the art on EndoVis2017, and is on par with state of the art on EndoVis2018, attesting its good generalizability. Note that categories between EndoVis2017 and EndoVis2018 do not fully overlap, the exceptions are the VS and SI categories. Despite this, we achieve very competitive results on them, showcasing the potential capability of our method for open-set text promptable surgical instrument segmentation.

### 4.4 Ablation Study

We conduct the ablation study on the EndoVis2017 dataset using images of sizes $448 \times 448$ and Ch_IoU and ISI_IoU as the evaluative metrics.

**Image encoder: multi-scale feature augmentation.** We ablate the proposed multi-scale feature augmentation (MSFA) in our method and offer the result in Table 3. A clear performance drop can be observed for ours w/o MSFA. MSFA enhances the contextual information for feature representation, thus improving the segmentation performance.

**Text encoder: global feature from [CLS] token.** We offer an variant by averaging the features from individual word tokens to produce the textural feature $F_T$. The result is in Table 3: one can see that [CLS] $\rightarrow$ Words decreases Ch_IoU and ISI_IoU scores by 0.46% and 0.48% respectively. This suggests that the [CLS] token can better encapsulate the global information of the text description.

Table 3: Ablation study for image and text encoder.

| Encoder | Ch_IoU | ISI_IoU |
|---|---|---|
| Ours | 77.79 | 76.45 |
| Ours w/o MSFA | 75.31 | 74.03 |
| Ours ([CLS] → Words) | 77.33 | 75.97 |

Table 4: Ablation study for text promptable mask decoder.

| Decoder | Ch_IoU | ISI_IoU |
|---|---|---|
| Ours | 77.79 | 76.45 |
| Ours w/ AP | 76.46 | 75.28 |
| Ours w/ CP | 42.38 | 37.61 |

Table 5: Ablation study for mixture of prompts; cls - class, tem - template.

| MoP | Ch_IoU | ISI_IoU |
|---|---|---|
| Ours w/ P-cls | 75.67 | 74.34 |
| Ours w/ P-tem | 75.86 | 74.39 |
| Ours w/ P-GPT | 77.19 | 76.09 |
| Ours w/ P-cls & tem | 76.01 | 74.53 |
| Ours w/ P-cls & tem & GPT | 77.79 | 76.45 |
| Ours (P-cls → P-Bard) | 77.76 | 76.44 |

Table 6: Ablation study for hard instrument area reinforcement.

| HIAR | Ch_IoU | ISI_IoU |
|---|---|---|
| Ours | 77.79 | 76.45 |
| Ours w/o HIAR | 75.22 | 73.98 |
| Ours w/o HAM | 75.98 | 74.57 |
| $r_t = 0.1$ | 77.03 | 76.01 |
| $r_t = 0.25$ | 77.79 | 76.45 |
| $r_t = 0.50$ | 77.57 | 76.32 |

Table 7: Ablation study for pixel-wise weight maps in mixture of prompts.

| Gating network | Ch_IoU | ISI_IoU |
|---|---|---|
| Pixel-wise | 77.79 | 76.45 |
| Image-wise | 76.80 | 75.77 |

Table 8: Ablation study for mask ratio threshold $r_t$ in hard instrument area reinforcement.

| mask ratio $r_t$ | Ch_IoU | ISI_IoU |
|---|---|---|
| 0.25 | 77.79 | 76.45 |
| 0.75 | 76.92 | 75.65 |

**Text promptable mask decoder: prompting schemes.** We study the importance of the two prompting schemes in Table 4. Results indicate that using only attention-based prompting (Ours w/ AP) outperforms using only convolutional-based prompting (Ours w/ CP) in terms of Ch_IoU and ISI_IoU scores. Moreover, employing both prompting schemes (Ours) sequentially improves the segmentation performance, suggesting that convolution-based prompting can further refine the output of the attention-based approach.

**Mixture of prompts: different prompts.** We offer the ablation study of MoP in Table 5. Results reveal that 1) using a single and simple prompt (*e.g.* Ours w/ P-cls) already performs very well, outperforming the state of the art in Table 1; 2) the proposed mixture of prompts can further improve the segmentation performance. Over the three prompt types, the one from GPT-4 performs the best, owing to its more comprehensive coverage of the knowledge for certain instrument. We have also tried to replace the simple class prompt with prompt generated from another large language model, Bard [13]. Yet, we do not observe significant performance difference, cf. Ours (P-cls → P-Bard). We suspect the reason is that the knowledge contained in the GPT-4 prompt and the Bard prompt highly overlap, given the input token length limit. Note that prompts generated by GPT-4 might slightly vary given different time stamps. We have tried to study this factor, yet find very trivial impact of it.

**Mixture of prompts: pixel-wise weight maps.** To confirm the need for pixel-wise weight maps in the proposed visual-textual gating network, we develop a simpler version by using the image-wise scalar weight following the design in MoE [6]. These scalar weights are normalized via softmax. Experiments in Table 7 show our pixel-wise weighting scheme is better than the image-wise variant.

**Hard instrument area reinforcement: modules and parameters.** If we remove this module from our method (Ours w/o HIAR), as shown in Table 6, the performance clearly drops (*e.g.* 2.57% and 2.47%). Next, we highlight the efficacy of the hard area mining (HAM) by removing it from HIAR, *i.e.* Ours w/o HAM; the HIAR module is then rather similar to a typical MAE module. We also observe a clear performance drop, underlining the crucial role of HAM in HIAR. Finally, we vary the making ratio threshold $r_t$ over 0.1, 0.25, and 0.5 in Table 6: we can see $r_t = 0.25$ works the best.

**Hard instrument area reinforcement: mask ratio $r_t$.** MAE [15] originally employs a high masking ratio (0.75), but in our study, we argue that a lower threshold (0.25) is more suitable: surgical instruments occupy a relatively small portion of the image, a low masking ratio will make the model concentrating on the hard area. Table 8 reveals a performance decrease with a masking ratio threshold of 0.75 when compared to our default setting of 0.25.

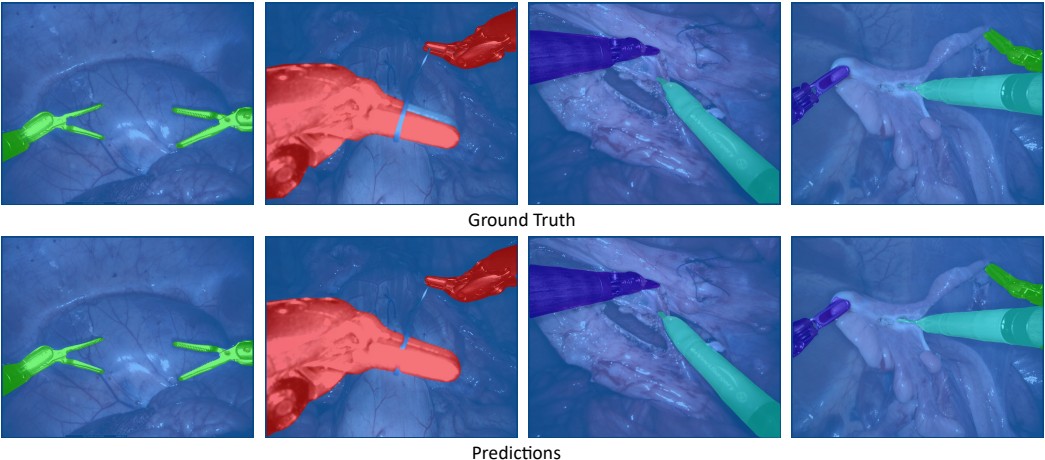

Figure 3: Our segmentation visualization uses distinct colors to represent surgical instrument categories, with ground truth and predictions displayed in the first and second rows, respectively.

## 4.5 Computational Analysis

We perform experiments to assess computational complexity and inference speed in Tab. 9. This is achieved by evaluating floating point operations per second (FLOPs) and frames per second (FPS) using a single A100 GPU. We run the experiment on EndoVis2017 [1] by resizing the input image to the default input sizes corresponding to different methods for testing (e.g. $800 \times 800$ for ISINet, $224 \times 224$ for MATIS, $416 \times 416$ for CRIS, $448 \times 448$ for CLIPSeg and Ours). In Tab. 9, it is evident that our model's computational complexity (FLOPs) and inference speed (FPS) aligns with other adapted text-promptable approaches (*i.e.* , CRIS [46] and CLIPSeg [29]), achieving real-time performance suitable for clinical applications. Compared to conventional segmentation methods (*i.e.* , ISINet [12] and MATIS [3]), ours appears to be clearly more efficient than ISINet; while it is marginally slower than MATIS, likely due to MATIS having lower complexity (FLOPs).

Table 9: Computational analysis between ours and representative works on the EndoVis2017.

| Method | FLOPs (G) | FPS |
|---|---|---|
| ISINet [12] | 264 | 19 |
| MATIS [3] | 66 | 27 |
| CRIS [46] | 196 | 19 |
| CLIPSeg [29] | 127 | 23 |
| **Ours (448)** | 125 | 22 |

## 5 Conclusion

In this paper, we introduce a novel approach for text promptable surgical instrument segmentation leveraging pretrained vision-language model. Our method addresses the challenge of instrument variety by redefining the task as a text promptable segmentation problem, which enhances the model's adaptability to new surgical instruments. Our promptable mask decoder carefully segments the surgical instruments in images via an attention-based and convolution-based prompting schemes. Our mixture of prompts mechanism effectively employs diverse prompts for better segmentation performance. While our hard instrument area reinforcement module intertwined with hard area mining significantly improves the segmentation performance on challenging area. Experimental results demonstrate our model's superior performance over state of the art. Future work will aim to further exploit the potential of this method for real-world surgical scenario.

## 6 Social Impact

Automated surgical instrument segmentation offers benefits to patients, surgeons, manufacturers, and society. Precise instrument tracking boosts surgical safety and precision, mitigating unintended tissue damage. Such automation eases surgeons' tasks, letting them concentrate on intricate procedures, and augments training for novices. Manufacturers integrating this technology could produce smarter surgical tools, aligning with AI-driven healthcare innovations to elevate patient care.

## Acknowledgment

This work was supported by King's Cambridge 1 Access Fund, Tongji Fundamental Research Funds for the Central Universities, and core funding from the Wellcome/EPSRC [WT203148/Z/16/Z; NS/A000049/1]. Computing resources provided by King's Computational Research, Engineering and Technology Environment (CREATE). For the purpose of open access, the authors have applied a CC BY public copyright licence to any Author Accepted Manuscript version arising from this submission. TV is supported by a Medtronic / RAEng Research Chair [RCSRF1819\7\34], and is co-founder and shareholder of Hypervision Surgical. MW and OA are supported by the EPSRC CDT [EP/S022104/1].

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
