# Supplementary Material for
# Text Promptable Surgical Instrument Segmentation with Vision-Language Models

**Zijian Zhou**[1]    **Oluwatosin Alabi**[2]    **Meng Wei**[2]    **Tom Vercauteren**[2]    **Miaojing Shi**[3⊠]

[1]Department of Informatics, King's College London
[2]School of Biomedical Engineering & Imaging Sciences, King's College London
[3]College of Electronic and Information Engineering, Tongji University
{first_name}.{last_name}@kcl.ac.uk; mshi@tongji.edu.cn

## 1    Appendix

In the supplementary material, we include additional method details, experimental results and analysis, and visualizations that could not be accommodated in the main text due to space constraints.

### 1.1    Prompts Generated by GPT-4 and Bard.

Below, we provide the surgical instrument prompts generated by utilizing OpenAI GPT-4 [8] and Google Bard [2]. They are used in our experiments section.

**OpenAI GPT-4 based prompts**    The input template for OpenAI GPT-4 is defined as: *Please describe the appearance of [class_name] in endoscopic surgery, and change the description to a phrase with subject, and not use colons.* We obtain the following prompts for different surgical instruments:

- Bipolar forceps. *Bipolar forceps have a slim, elongated tweezer-like design with opposing tips, are silver-colored, made from high-quality metal, and feature an insulated shaft for controlled energy application.*

- Prograsp forceps. *Prograsp forceps possess curved scissor-like handles, specialized grasping tips with interlocking jaws, a ratcheting mechanism, and color-coded markings for easy identification during surgery.*

- Large needle driver. *Large needle drivers feature elongated handles, sturdy gripping surfaces, a curved or straight jaw tip for securely holding needles, and a locking mechanism to ensure precision and control.*

- Vessel sealer. *Vessel sealers have elongated handles, scissor-like controls, and specialized jaws with a combination of sealing and cutting surfaces, designed for securely sealing and dividing blood vessels and tissue bundles.*

- Grasping retractor. *Grasping retractors display elongated shafts, curved or straight jaws with serrated or smooth surfaces for gripping tissues, and a handle mechanism for precise control and retraction of the target area.*

- Monopolar curved scissors. *Monopolar curved scissors showcase elongated handles, curved cutting edges for precise dissection, and an insulated shaft, allowing controlled application of electrical energy for cutting and coagulation.*

---

⊠ Corresponding author.

37th Conference on Neural Information Processing Systems (NeurIPS 2023).

- Ultrasound probe. *Ultrasound probes feature a long, slender handle, a small transducer head for producing ultrasound waves, and a flexible cable connecting the probe to the ultrasound machine for real-time imaging guidance.*
- Suction instrument. *Suction instruments appear as elongated tubes with a narrow, hollow tip for fluid and debris removal, connected to a handle and tubing system for vacuum generation and precise control during the procedure.*
- Clip applier. *Clip appliers feature elongated handles, a shaft with a specialized tip for holding and releasing clips, and a mechanism to advance and deploy the clips precisely for secure tissue or vessel closure.*

**Google Bard based prompts**   The input template for Google Bard is defined as: *Please describe the appearance of [class_name] in endoscopic surgery, and condense it into a short sentence with a length of about 50 words.* We obtain the following prompts for different surgical instruments:

- Bipolar forceps. *Bipolar forceps used in endoscopic surgery are insulated stainless steel or titanium forceps with two metal disks on the tips of the jaws that conduct electrical current when closed.*
- Prograsp forceps. *Prograsp forceps are long, thin surgical instruments with two jaws that are used to grasp tissue during endoscopic surgery.*
- Large needle driver. *Large needle driver in endoscopic surgery is a long, slender instrument with serrated jaws for grasping and manipulating needles and sutures.*
- Vessel sealer. *Vessel sealer in endoscopic surgery is a handheld device that uses heat or energy to seal and/or cut blood vessels during surgery.*
- Grasping retractor. *Grasping retractor is a long, thin, and flexible instrument with toothed jaws that is used to hold tissue in place during endoscopic surgery.*
- Monopolar curved scissors. *Monopolar curved scissors are a type of surgical scissors used in minimally invasive procedures to cut tissue.*
- Ultrasound probe. *Ultrasound probe is a thin, flexible tube with a camera and ultrasound transducer at the tip.*
- Suction instrument. *Suction instrument in endoscopic surgery is a thin, tube-like device used to remove fluids and debris.*
- Clip applier. *Clip applier is a handheld device with two handles and a shaft that is inserted into the body through a small incision, which is loaded with titanium clips that are used to close blood vessels or other openings.*

## 1.2   Experiments on More Datasets

To further validate our method, we conduct experiments on the EndoVis2019 [9] and Cholecseg8k [4] datasets. Below, we introduce the EndoVis2019 and Cholecseg8k datasets, describe the evaluation metrics for each dataset, and present the experimental results.

**Datasets.** *EndoVis2019* [9] is derived from 30 minimally invasive surgical procedures, including 10 rectal resection, 10 proctocolectomy, and 10 sigmoid resection procedures. A total of 10,040 images are extracted from these procedures. The dataset consists of both training and test cases. Each case contains a 10-second video snippet with 250 endoscopic image frames and a reference annotation for the last frame. *Cholecseg8k* [4] contains 80 videos of cholecystectomy surgeries performed by 13 surgeons. Each video is recorded at 25 FPS and has annotations for instruments and operation phases. Each video clip contributes 80 image frames, and for each of these frames, the dataset includes raw image data, annotations, and colour masks. In total, the dataset comprises 101 directories with a collection of 8,080 frames.

**Metrics.** For *EndoVis2019*, consistent with the competition's evaluation protocol [9], we use the Dice Similarity Coefficient (DSC) and Normalized Surface Dice (NSD) to assess the segmentation performance. For *Cholecseg8k*, following the protocols from SP-TCN [3], we split the dataset into training and testing sets (videos 12, 20, 48 and 55 for testing and others for training) and utilize the mean Intersection over Union (mean IoU) as the evaluation metric.

**Results.** For *EndoVis2019*, the results are shown in Tab. 1, our method (input size 448) notably surpasses the competition's top performers, with +3% increase in DSC and +2% enhancement in NSD, which demonstrates the superiority of our method. It's worth noting that our approach is designed for multi-class segmentation while is tested for binary class segmentation. Despite this, the performance improvement by ours over SOTA underscores its efficacy.

For *Cholecseg8k*, the results are shown in Tab. 2, the mean IoU of our method is 71.03%. It's evident that our method surpasses the current SOTA by 1.65% in mean IoU, even though SP-TCN leverages temporal information from videos to boost the performance, while our method solely relies on individual image data. It's worth noting that for the Cholecseg8k, we use the same prompt generation method described in our paper to obtain prompts for both tissues and instruments. The result demonstrates that prompts for tissues are appropriately generated following our method, further attesting to the generalizability of our method.

Table 1: Comparison between our method and other state-of-the-art methods on the EndoVis2019 dataset.

| Method | DSC | NSD |
|---|---|---|
| haoyun [9] | 0.89 | 0.89 |
| CASIA-SRL [9] | 0.78 | 0.89 |
| **Ours (448)** | 0.92 | 0.91 |

Table 2: Comparison between our method and other state-of-the-art methods on the Cholecseg8k dataset.

| Method | mean IoU |
|---|---|
| Swin base [6] | 68.42 |
| Swin base + SP-TCN [3] | 69.38 |
| **Ours (448)** | 71.03 |

## 1.3 Comparison to SAM.

Whilst we were working on our work, the segment anything model (SAM) [5] was released concurrently. SAM possesses the capability to output corresponding object masks through visual or text prompts. Its officially released implementation however only contains visual prompts (point, box and mask), lacks text prompting. We thus leverage a community-based solution, lang-segment-anything [7], which enables this function for SAM. We let SAM use the same set of prompts as we do and present the results in Tab. 1.3. We found that this SAM variant struggles with medical prompts, performing significantly inferior than ours. This suggests challenges with medical concepts without fine-tuning. Additionally, we notice that the text encoder's output in this unofficial implementation of SAM might not align well with its visual encoder's output, potentially leading to decreased performance.

Table 3: Comparison to SAM on EndoVis2017 and EndoVis2018.

| Method | EndoVis2017 | | | EndoVis2018 | | |
|---|---|---|---|---|---|---|
| | Ch_IoU | ISI_IoU | mc_IoU | Ch_IoU | ISI_IoU | mc_IoU |
| SAM-variant [7] | 17.77 | 14.32 | 10.28 | 22.08 | 17.88 | 12.33 |
| **Ours (448)** | 77.79 | 76.45 | 54.78 | 82.67 | 81.54 | 65.48 |

## 1.4 Analysis of Hard Instrument Area Reinforcement

We conducted the analysis of the model's identified hard instrument area aligns with the surgeons' perspective. In Fig. 1, we visualize the hard instrument areas and observe that these areas predominantly reside at the instrument's clasper and shaft positions (red rectangles). For the clasper, due to its deep interaction with the tissue, making the image complex and the segmentation challenging. The shaft, on the other hand, presents issues because different instruments often have similar shaft appearances, leading to model misinterpretations. After consulting with surgeons, they agreed that the clasper is the hard area to identify, aligning with our findings, but they didn't find the shaft as challenging. The difference arises because the instrument clasper, influenced by factors like lighting, can more easily be mistaken for tissue. While for surgeons, the classification of the instrument shaft is inferred from the clasper, making the shaft a non-challenging area for them. However, for models, grasping the relationship between the clasper and shaft might not be as intuitive, leading to misclassification. This observation suggests that future work should focus on modeling the relationship between the clasper and shaft to enhance segmentation performance across different parts.

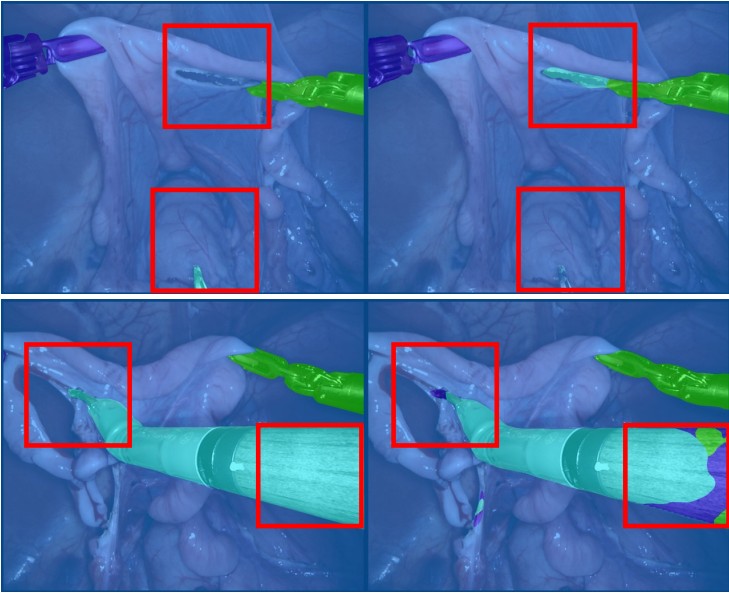

Figure 1: Illustration of the hard area generated in hard instrument area reinforcement. For each of the two examples (top, bottom), the left side displays the segmentation ground truth, while the right side is segmentation prediction, the red rectangle is where the hard area is concentrated.

## 1.5 More Visualization Results

We introduce a mixture of prompts mechanism (MoP) for fusing results from multiple prompts. Visualization results for various instruments using MoP are given in Figures 2 through 6. Our method distinguishes instrument categories correctly and demonstrates superior segmentation performance, particularly at subtle details.

In Figure 7 and Figure 8, we also compare our method with the visualization results of ISINet [1] and CRIS [10]. Compared with ISINet, our method achieves better classification accuracy for different instruments. Compared with the CRIS, our method is more accurate on edge segmentation.

Prompts:
- $T_1$: bipolar forceps.
- $T_2$: the surgical instrument area represented by the bipolar forceps.
- $T_3$: bipolar forceps have a slim, elongated tweezer-like design with opposing tips, are silver-colored, made from high-quality metal, and feature an insulated shaft for controlled energy application.

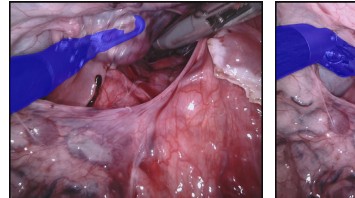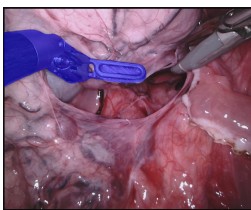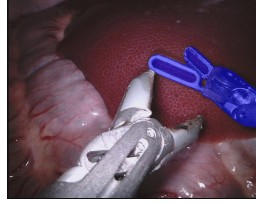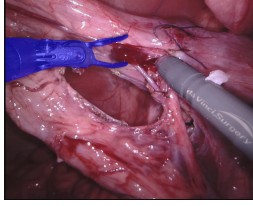

Figure 2: Visualization results for bipolar forceps. Top: the prompts signifying bipolar forceps, namely $T_1, T_2, T_3$, are concurrently fed into the model. Bottom: the visualizations represent the outputs derived from these prompts across diverse images.

Prompts:
- T$_1$: prograsp forceps.
- T$_2$: the surgical instrument area represented by the prograsp forceps.
- T$_3$: prograsp forceps possess curved scissor-like handles, specialized grasping tips with interlocking jaws, a ratcheting mechanism, and color-coded markings for easy identification during surgery.

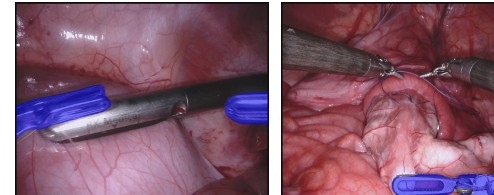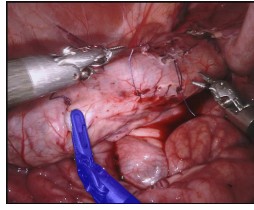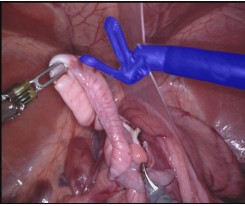

Figure 3: Visualization results for prograsp forceps. Top: the prompts signifying prograsp forceps, namely $T_1, T_2, T_3$, are concurrently fed into the model. Bottom: the visualizations represent the outputs derived from these prompts across diverse images.

Prompts:
- T$_1$: large needle driver.
- T$_2$: the surgical instrument area represented by the large needle driver.
- T$_3$: large needle drivers feature elongated handles, sturdy gripping surfaces, a curved or straight jaw tip for securely holding needles, and a locking mechanism to ensure precision and control.

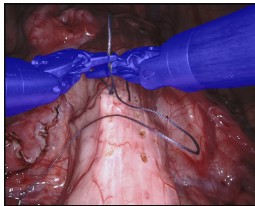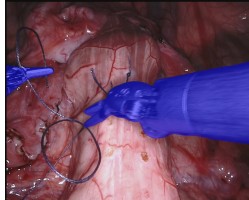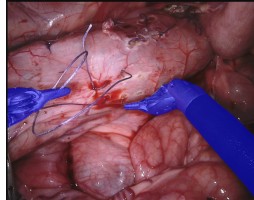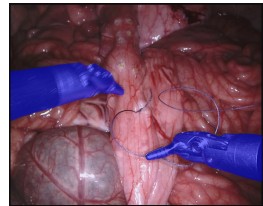

Figure 4: Visualization results for large needle driver. Top: the prompts signifying large needle driver, namely $T_1, T_2, T_3$, are concurrently fed into the model. Bottom: the visualizations represent the outputs derived from these prompts across diverse images.

Prompts:
- T$_1$: vessel sealer.
- T$_2$: the surgical instrument area represented by the vessel sealer.
- T$_3$: vessel sealers have elongated handles, scissor-like controls, and specialized jaws with a combination of sealing and cutting surfaces, designed for securely sealing and dividing blood vessels and tissue bundles.

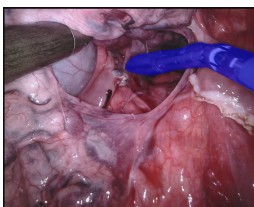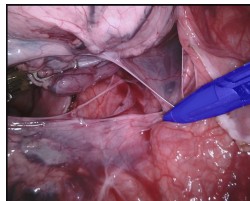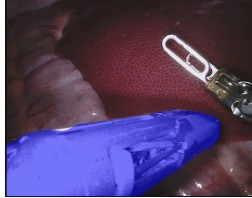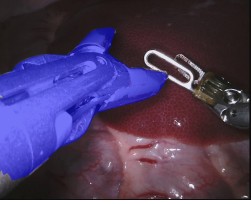

Figure 5: Visualization results for vessel sealer. Top: the prompts signifying vessel sealer, namely $T_1, T_2, T_3$, are concurrently fed into the model. Bottom: the visualizations represent the outputs derived from these prompts across diverse images.

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

Prompts:
- $T_1$: monopolar curved scissors.
- $T_2$: the surgical instrument area represented by the monopolar curved scissors.
- $T_3$: monopolar curved scissors showcase elongated handles, curved cutting edges for precise dissection, and an insulated shaft, allowing controlled application of electrical energy for cutting and coagulation.

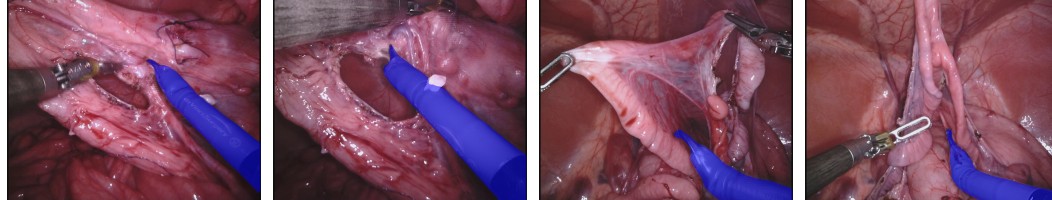

Figure 6: Visualization results for monopolar curved scissors. Top: the prompts signifying monopolar curved scissors, namely $T_1, T_2, T_3$, are concurrently fed into the model. Bottom: the visualizations represent the outputs derived from these prompts across diverse images.

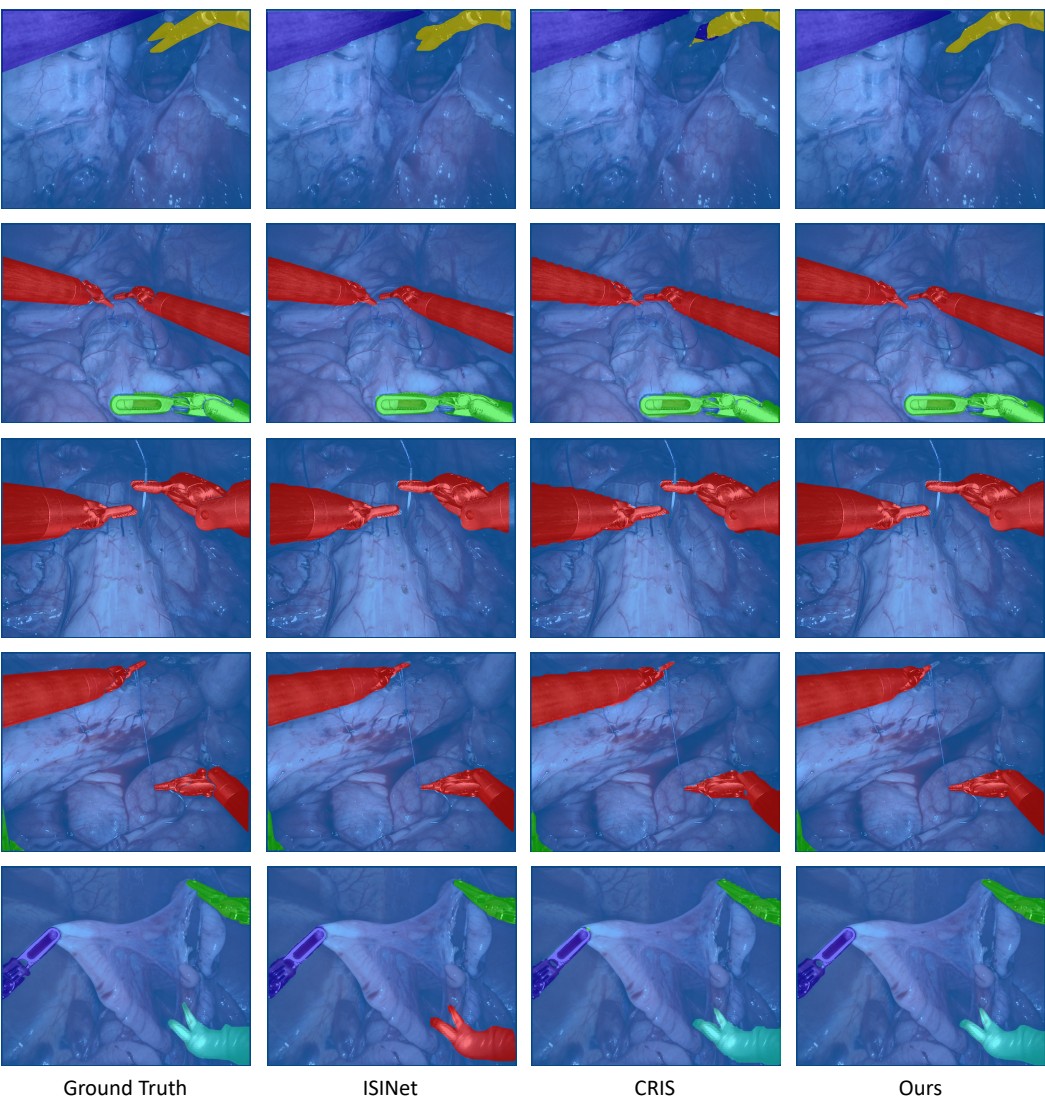

| Ground Truth | ISINet | CRIS | Ours |

Figure 7: Comparison of visualization results of different methods.

[4] W-Y Hong, C-L Kao, Y-H Kuo, J-R Wang, W-L Chang, and C-S Shih. Cholecseg8k: a semantic segmentation dataset for laparoscopic cholecystectomy based on cholec80. *arXiv preprint*, 2020.

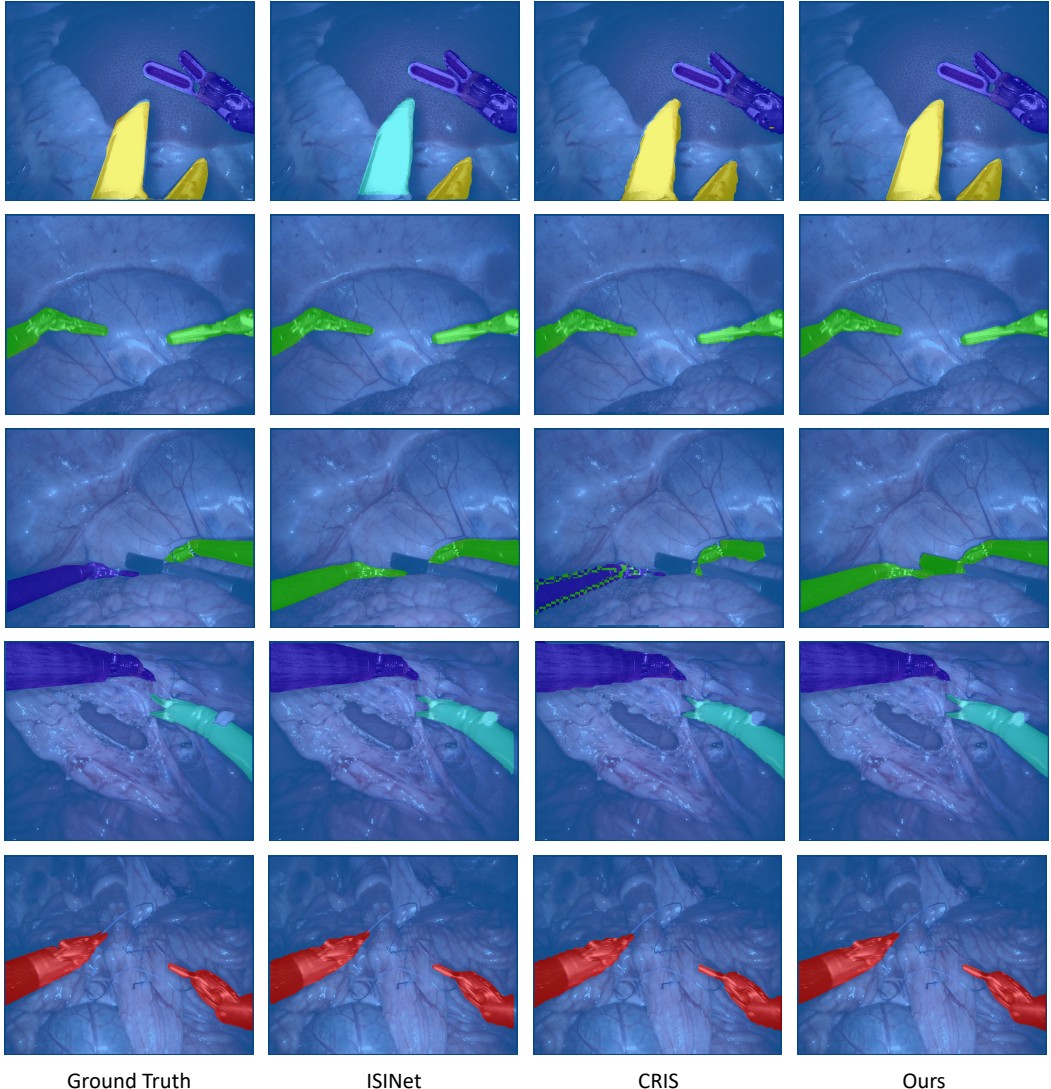

| Ground Truth | ISINet | CRIS | Ours |

Figure 8: Comparison of visualization results of different methods.

[5] Alexander Kirillov, Eric Mintun, Nikhila Ravi, Hanzi Mao, Chloe Rolland, Laura Gustafson, Tete Xiao, Spencer Whitehead, Alexander C Berg, Wan-Yen Lo, et al. Segment anything. *arXiv preprint*, 2023.

[6] Ze Liu, Yutong Lin, Yue Cao, Han Hu, Yixuan Wei, Zheng Zhang, Stephen Lin, and Baining Guo. Swin transformer: Hierarchical vision transformer using shifted windows. In *ICCV*, 2021.

[7] Luca Medeiros. Language segment-anything, 2023. URL https://github.com/luca-medeiros/lang-segment-anything.

[8] OpenAI. GPT-4 technical report. *arXiv preprint*, 2023.

[9] Tobias Ross, Annika Reinke, Peter M Full, Martin Wagner, Hannes Kenngott, Martin Apitz, Hellena Hempe, Diana Mindroc Filimon, Patrick Scholz, Thuy Nuong Tran, et al. Robust medical instrument segmentation challenge 2019. *arXiv preprint*, 2020.

[10] Zhaoqing Wang, Yu Lu, Qiang Li, Xunqiang Tao, Yandong Guo, Mingming Gong, and Tongliang Liu. Cris: Clip-driven referring image segmentation. In *CVPR*, 2022.