# OpenReview forum: "Text Promptable Surgical Instrument Segmentation with Vision-Language Models"
_NeurIPS.cc/2023/Conference — NeurIPS 2023 poster_

### Official Review · Reviewer_fFdZ · 2023-07-01

**Soundness:** 3 good
**Presentation:** 3 good
**Contribution:** 2 fair
**Rating:** 5
**Confidence:** 4

**Summary:**

This manuscript presents a clip-assisted semantic image segmentation method for surgical instruments. In terms of methodology, the proposed work can be viewed as an adaption of CLIPSeg to a domain-specific problem. Compared with CLIPSeg, a mixture of prompt strategy is used for augmenting text prompt information. Hard sample mining is also employed to further improve segmentation performance. The image segmentation network architecture has also been optimized. The proposed method is evaluated on two public surgical instrument segmentation datasets and is shown to outperform some fully-supervised and clip-assisted methods.

**Commented after rebuttal**: I would like to thank the authors, other reviewers, and the ACs. I have carefully read through the rebuttals, and the comments from other reviewers. The discussion with the authors is constructive. Most of my raised concerns have been properly addressed.

**Strengths:**

The proposed method yields improved performance on two surgical instrument segmentation datasets, compared with some existing fully-supervised/clip-assisted methods.

Employing text prompt brings additional flexibility in terms of the categories of segmentation targets. This is a major advantage of clip-assisted segmentation over conventional fully-supervised segmentation.

The proposed method is introduced in sufficient detail and can be easy to follow.

**Weaknesses:**

The take-home information for readers may be limited/unclear. Despite that domain-specific adaptions on top of vanilla CLIPSeg leads to improved performance on surgical instrument segmentation, these contributions themselves may not be of sufficient interest to readers: hard sample mining, feature pyramid, generating multiple text prompts are already the common practice and should have been well-known for the community. The authors are encouraged to highlight their key take-home information to readers, and argue how the proposed work brings new knowledge.

In the abstract and introduction, the authors highlight improved flexibility when dealing with new segmentation targets of clip-assisted segmentation models. However, this point is not sufficiently evaluated and discussed in the experiments: how does CLIPSeg and CRIS that are also clip-driven work in this context?

The authors may also want to discuss/comment on a closely-related existing work [1].

1. CLIP-Driven Universal Model for Organ Segmentation and Tumor Detection

**Questions:**

Judging by the ablation study (table 3 - 6), the contributions of individual components, except for the text encoder, seem to be quite marginal (the ablated models seem to be already quite strong). Therefore, from the authors' point of view, which factor accounts for most of the improved performance over CLIPSeg and CRIS? Or if there are other components that make a difference? Does the proposed method employ a stronger segmentation backbone or better training technique? The authors are encouraged to provide more details about the compared methods to avoid confusion.

From the authors' perspective, what is the core take-home information to the readers?

**Limitations:**

Automated instrument image segmentation may affect the outcome of a surgery, which may lead to real-world impact to the clinicians and the patient. The authors are encouraged to discuss the potential impact of their work for patients, clinicians, device manufacturers, and the society.

The authors are also encouraged to discuss if the proposed method exacerbate/mitigate potential risks and biases in image segmentation.

---

> ### Author Rebuttal · Authors · 2023-08-09
>
> ### [fFdZ-W1, Q2] Take-home information for readers.
>
> To our knowledge, we're the first to introduce text promptable method in surgical instrument segmentation.
> Through problem-driven thinking, we proposed Mixture of Prompts (MoP) and Hard Instrument Area Reinforcement (HIAR) modules tailored for specific challenges in this field.
> MoP addresses instrument similarity by integrating detailed text descriptions, hence enhancing the classification  among of instruments.
> HIAR improves segmentation in areas where instruments and tissues often overlap.
> Both MoP and HIAR are inspired by advances in computer vision segmentation, but the concrete ideas of mixing the strength of multiple prompts and reinforcing hard regions in a masked auto-encoder are actually novel in the computer vision literature.
> Finally, while designed for this specific task, we believe our method's applicability also extends to other segmentation tasks, underscoring its efficacy.
>
> ### [fFdZ-W2] The performance of CRIS and CLIPSeg for the cross dataset validation.
>
> For comprehensive analysis, we've added comparisons against CRIS and CLIPSeg by cross dataset validation in following tables.
> In the following tables, it can be seen that CRIS and CLIPSeg exhibit a significant drop in performance in cross-dataset validation compared to their corresponding counterparts and ours, indicating weaker generalization capabilities for both methods.
>
> Cross dataset validation for CRIS and CLIPSeg on EndoVis2017:
>  Method           | Ch_IoU | ISI_IoU | mc_IoU
> ------------------|--------|---------|--------
>  CRIS             | 69.94  | 67.83   | 38.21
>  Cross-CRIS       | 61.33  | 59.87   | 31.52
>  CLIPSeg          | 70.15  | 65.02   | 33.42
>  Cross-CLIPSeg    | 60.73  | 57.26   | 28.71
>  Ours       | 79.90  | 77.83   | 56.22
>  Cross-Ours | 72.18  | 70.44   | 49.09
>
> Cross dataset validation for CRIS and CLIPSeg on EndoVis2018:
>  Method           | Ch_IoU | ISI_IoU | mc_IoU
> ------------------|--------|---------|--------
>  CRIS             | 74.10  | 72.29   | 46.04
>  Cross-CRIS       | 53.91  | 52.18   | 29.46
>  CLIPSeg          | 74.95  | 69.86   | 39.70
>  Cross-CLIPSeg    | 51.35  | 50.77   | 27.62
>  Ours       | 84.92  | 83.61   | 65.44
>  Cross-Ours | 66.25  | 64.92   | 37.34
>
> ### [fFdZ-W3] Discuss the Universal Model.
> Thank you for pointing out the "Universal Model" paper.
> Our work primarily focuses on surgical instrument segmentation, while the Universal Model targets organ segmentation and tumor detection, thus differing in their research domains.
> This is why it was not initially cited or discussed in our manuscript.
> However, consistent with our approach, the paper also leverages the CLIP vision-language model to enhance the performance of medical image segmentation models.
> We will cite this work in our paper and discuss it in the appropriate sections.
>
> ### [fFdZ-Q1] More explanation for ablation study.
>
> In Tabs. 3-6 in the paper, we present ablation study results.
> From our final model, we remove each single module to observe performance changes.
> Notably, without multi-scale feature augmentation (MSFA), there's a 2.4% performance drop; without Mixture of Prompts (MoP), the drop is 2.1%; and without Hard Instrument Area Reinforcement (HIAR), it's 2.5%.
> These aren't marginal but significant if we compare the performance increase between the previous two SOTA (S3Net [5] and MATIS [3] in Tab. 1 of the paper).
>
> Moreover, our modules are complementary, omitting both MoP and HIAR leads to a 3.9% drop.
> Removing MSFA, MoP, and HIAR all together results in a 5.2% decrease; this, as our baseline, is still 2.6% higher than CRIS (Ch_IoU result in Tab. 1 in our paper).
> It is because our baseline benefits from data augmentation strategies (random crop, horizon flip, random rotate) and employs a ViT-based CLIP model trained on Laion2B.
>
> We will elaborate on these details in the revised paper to avoid ambiguity.
>
> ### [fFdZ-L1] The potential impact for patients, clinicians, device manufacturers, and the society.
>
> Enhanced automated surgical instrument segmentation offers manifold benefits to patients, surgeons, manufacturers, and society.
> Precise instrument tracking boosts surgical safety and precision, mitigating unintended tissue damage.
> Such automation eases surgeons' tasks, letting them concentrate on intricate procedures, and augments training for novices.
> Manufacturers integrating this technology could produce smarter surgical tools, aligning with AI-driven healthcare innovations to elevate patient care.
>
> ### [fFdZ-L2] If the proposed method exacerbates/mitigates potential risks and biases in image segmentation.
>
> Model bias due to non-representative distributions of gender or race in the training data does not apply to instrument segmentation datasets.
> Nevertheless, there could still be a human-driven bias in surgical instrument segmentation training data due to surgeons' preference for specific surgical instruments in certain scenarios over other tools.
> This bias can be mitigated by our text-promptable approach, which allows for using text prompts that are aimed at mitigating these scenarios (e.g. providing specific surgical scenario/function descriptions for certain surgical instrument).
>
> The study into the risk and biases of our method differs from this research's current focus; hence we leave further investigations for future works.

---

> > ### Comment · Reviewer_fFdZ · 2023-08-15
> >
> > Hi authors,
> >
> > Thank you so much for the clarification. Most of my concerns are properly addressed and I am very pleased to see the proposed method yielding improved performance compared with existing prompt-driven methods with large margins on both IID and OOD settings. I am also pleased to see the in-depth analysis about the potential impacts and biases, which comprises responsible research and innovation.
> >
> > Just two minor comments:
> >
> > 1. I am still a bit missing about the core take-home information. Despite the authors' claim to be the first to apply prompt-driven segmentation to surgical instrument segmentation, unless the drastic differences between segmenting surgical instruments versus segmenting radiological images/RGB images are adequately explained, I would not be 100% convinced that itself comprises a major contribution in the context of NeurIPS.
> >
> > 2. Still, the improved segmentation performance attribute to a synergy among multiple components, instead of the prompt-driven mechanism alone. The authors are expected to make this point very clear to readers, and emphasize/argue a) why a synergy among these components is crucial for the targeted task; b) why the proposed synergy is applicable to other domains that are relevant to the readers of NeurIPS (e.g., segmenting RGB and radiological images).

---

> > > ### Author Response · Authors · 2023-08-16
> > >
> > > Thank you very much for acknowledging our rebuttal and paper.
> > > Below, we provide answers to your newly raised comments:
> > >
> > > ## Q1
> > >
> > > Thanks for your question. Below we summarize the challenges lying in surgical images:
> > >
> > > - In surgical images, it is common that instruments may exhibit significant similarity, necessitating the model that can distinguish their subtle differences for accurate segmentation.
> > >
> > > - As surgeries progress, surgical instruments might cut, suture, or otherwise manipulate tissues, altering their shapes. This can lead to tissues obscuring the instruments and potentially making some parts of the tissues even resemble the instruments.
> > >
> > > - In surgical settings, the use of diminutive endoscopic devices with small lenses inherently constrains the imaging quality due to hardware limitations.
> > >
> > > - The areas into which the endoscope is inserted, like the gastrointestinal tract, are continuously moving, which can add complexities to the segmentation of surgical instruments.
> > >
> > > Given above, on one hand, when comparing with those objects in natural scenes in RGB images which often have distinct and rigid boundaries, the continuous morphological changes of instruments and tissues during surgeries make instrument segmentation easily affected by tissue occlusions and variations in illumination, etc.
> > > On the other hand, when dealing with radiological images, different segmentation challenges emerge: the presence of various imaging modalities, intrinsic noise, limited contrast, and the potential for artefacts collectively introduce complexity to the precise segmentation in radiological images.
> > >
> > > Overall, we firmly believe that research on surgical instrument segmentation is both challenging and meaningful.
> > > Our methods have achieved significant improvements over text-promptable methods (e.g., CRIS, CLIPSeg) developed on natural images.
> > > We follow a problem-driven approach, in response to the aforementioned challenges in surgical images, we introduce the mixture of prompts module to address instrument similarity by integrating detailed text descriptions; moreover, the hard instrument area reinforcement module further amplifies the model's precise segmentation performance, especially in tricky regions where distinctions between instruments and tissues become blurred.
> > >
> > > Finally, although our method is designed for surgical instrument segmentation, it should also have the merits in natural image segmentation, especially when encountering challenges similar to those in surgical contexts.
> > > For instance, when undertaking fine-grained segmentation or in the presence of heavy occlusions. Therefore, our method holds potential for broader applications in computer vision tasks, and NeurIPS would be an excellent venue to showcase our work.
> > >
> > > ## Q2
> > >
> > > Thanks for this question.
> > >
> > > First, we emphasize that our method is not merely a synergy of a few components.
> > > For the text-promptable pipeline itself, there are inherent novelties instead of a simple adaptation from existing approaches. For instance, we devise a multi-scale fusion scheme for image encoder and the mask decoder integrated both attention-based and convolution-based prompting schemes to facilitate text features in guiding visual features for segmentation prediction (see Tab. 3 & 4 in our paper for their improvements).
> > >
> > > Building upon the text-promptable method, we introduce the mixture of prompts, substantially enhancing model performance, especially in the prediction of novel instruments (see Tab. 5 in our paper).
> > > Additionally, to overcome classification inaccuracies during segmentation, we incorporate the hard instrument area reinforcement module (see Tab. 6 in our paper).
> > > All these modules, as mentioned in the above answer, follow a problem-driven paradigm in the surgical instrument segmentation domain.
> > >
> > > Next, although our modules have been primarily validated in surgical instrument segmentation, we believe they can offer insights or potential benefits for segmentation tasks in other domains (e.g., RGB and radiological images), especially when facing challenges akin to those found in our surgical images.
> > > For example, in tasks such as fine-grained segmentation of natural images, our mixture of prompts module could be effectively employed and for scenarios with heavy occlusions, our hard instrument area reinforcement module could be particularly suited.
> > > Our approach exhibits strong adaptability, as demonstrated on the CholecSeg8k dataset where it not only segments instruments but also various tissues.
> > > Experimentally, our method surpasses the current SOTA in performance (see results in the global response).
> > >
> > > In summary, our method was borne out of a problem-driven necessity.
> > > To address these challenges, we introduce various modules which, when synergized, significantly enhance performance.
> > > We believe our approach harbors immense potential for broader applications within the computer vision domain, especially for methods centered on text prompts.

---

> > > > ### Comment · Reviewer_fFdZ · 2023-08-17
> > > >
> > > > Hi Authors,
> > > >
> > > > Thank you for your clarifications. I would encourage you to concisely emphasize your arguments above in the paper. These would help the readers to grasp the nature of your problem as well as the key contributions. (Still I am not fully convinced by response 2: I would argue performance improvement would also be seen when hard negative mining and feature pyramid were applied to ClipSeg.)

---

> > > > > ### Author Response · Authors · 2023-08-18
> > > > >
> > > > > Thanks a lot for your response, we will surely add the relevant content in rebuttal into the final version of our paper.
> > > > >
> > > > > Regarding the Hard Instrument Area Reinforcement (HIAR) module, we agree that improvement might be observed if it is applied to CLIPSeg.
> > > > > As being said, our method was designed in a problem-driven manner, integrating HIAR into CLIPSeg is likely to also improve the performance in surgical instrument segmentation domain.
> > > > > Besides, for similar scenarios in natural images, e.g. objects are occluded, integrating HIAR into CLIPSeg might also help.
> > > > > Overall, these potential improvements indeed show the generalizability of our proposed module as well as its merits beyond the surgical instrument segmentation task.

---

### Official Review · Reviewer_9oei · 2023-07-03

**Soundness:** 2 fair
**Presentation:** 3 good
**Contribution:** 2 fair
**Rating:** 5
**Confidence:** 4

**Summary:**

This paper proposes a novel text promptable surgical instrument segmentation approach to overcome challenges associated with the diversity and differentiation of surgical instruments by using the large CLIP model and a text promptable mask decoder. The experiments show the effectiveness of the proposed method on surgical instrument segmentation. However, I am very concerned with the novelty of the proposed modules due to the limited situations.

**Strengths:**

1, The paper is well-written and easy to understand.

2, The experiments show the effectiveness of the proposed modules.

**Weaknesses:**

1, There are too few novelties due to most of the proposed modules being explored in the traditional segmentation tasks.

2, The dataset is too limited to prove the effectiveness of the proposed modules. Furthermore, there is no specific module designed for surgical instrument segmentation.

3, Lacking the comparison of inference time.

**Questions:**

1, How about comparing the proposed method and the Segment Anything model?

2, What’s the difficulty of surgical instrument segmentation? I understand that there might be a lack of some segmentation datasets leading to overfitting.

---

> ### Author Rebuttal · Authors · 2023-08-09
>
> ### [9oei-W1, W2] Few novelties and modules already explored in traditional segmentation.
>
> We respectfully disagree.
> First, to our knowledge, we're the first to introduce text promptable method in surgical instrument segmentation.
> Second, through problem-driven thinking, we proposed the Mixture of Prompts (MoP) and Hard Instrument Area Reinforcement (HIAR) modules tailored for specific challenges in this field.
> MoP addresses instrument similarity by integrating detailed text descriptions, hence enhancing the classification among instruments.
> HIAR improves segmentation in areas where instruments and tissues often overlap.
> Both MoP and HIAR are inspired by advances in computer vision segmentation, but the concrete ideas of mixing the strength of multiple prompts and reinforcing hard regions in a masked auto-encoder are actually novel in the computer vision literature.
> Finally, while designed for this specific task, we believe our method's applicability also extends to other segmentation tasks, underscoring its efficacy.
>
> ### [9oei-W2] The dataset is too limited.
>
> It is a common practice to evaluate methods on the two established datasets, EndoVis 2017 and 2018 (see [36, 18, 11, 47, 5, 3]). To further validate our approach, we have added experimental results on EndoVis2019 and CholecSeg8k. Please refer to our global response section.
>
> ### [9oei-W3] Lacking the comparison of inference time.
>
> We assess the computational complexity and inference speed by evaluating the floating point operations per second (FLOPs) and frames per second (FPS) respectively, using a single A100 GPU.
> We run the experiment on EndoVis2017 by resizing the input image to the default input sizes corresponding to different methods for testing (e.g. $800 \times 800$ for ISINet, $224 \times 224$ for MATIS, $416 \times 416$ for CRIS, $448 \times 448$ for CLIPSeg and Ours).
> From the table below, it's evident that our model's computational complexity (FLOPs) and inference speed (FPS) align with other adapted text-promptable approaches (i.e., CRIS and CLIPSeg), achieving real-time performance suitable for clinical applications.
> On the other hand, compared to conventional segmentation methods (i.e., ISINet and MATIS), ours appears to be clearly more efficient than ISINet; while it is marginally slower than MATIS [3], likely due to MATIS's small input size.
>
>  Method       | FLOPs (G) | FPS
> --------------|-----------|-----
>  ISINet [11]  | 264       | 19
>  MATIS [3]    | 66        | 27
>  CRIS [39]    | 196       | 19
>  CLIPSeg [24] | 127       | 23
>  Ours         | 125       | 22
>
> ### [9oei-Q1] Compare with Segment Anything.
>
> We value the recommendation to compare our method with the Segment Anything model (SAM), though it is only recently appeared.
> While SAM is proficient with various input prompts, its official released implementation only contains visual prompts (point, box and mask), lacks text prompting.
> However, visual prompts differ from our task setting, we aim to evaluate the performance of text-promptable SAM.
> We leverage a community-based solution (lang-segment-anything from luca-medeiros), which enables this function for SAM.
> We let SAM use the same set of prompts as we do and present the results in following tables.
> We found that SAM struggles with medical prompts, performing significantly inferior than ours.
> This suggests challenges with medical concepts without fine-tuning.
> Additionally, we notice that the text encoder's output in this unofficial implementation of SAM might not align well with its visual encoder's output, potentially leading to decreased performance.
>
> SAM results on EndoVis2017:
>  Method     | Ch_IoU | ISI_IoU | mc_IoU
> ------------|--------|---------|--------
>  SAM       | 17.77   | 14.32    | 10.28
>  Ours (448) | 77.79  | 76.45   | 54.78
>
> SAM results on EndoVis2018:
>  Method     | Ch_IoU | ISI_IoU | mc_IoU
> ------------|--------|---------|--------
>  SAM        | 22.08  | 17.88    | 12.33
>  Ours (448) | 82.67  | 81.54   | 65.48
>
> ### [9oei-Q2] What’s the difficulty of surgical instrument segmentation? Will the lack of segmentation datasets lead to overfitting?
>
> As described in the introduction of the paper, surgical instrument segmentation faces two typical difficulties.
> The first is the continual emergence of new instruments, necessitating frequent model retraining to accommodate these novelties.
> The second pertains to misclassification during segmentation of similar instruments and their boundaries.
> Our paper proposes solutions tailored to these challenges: To address the emergence of new surgical instruments, we pioneer the definition of surgical instrument segmentation in a text-promptable format, enabling the open-set segmentation. We also introduce the Mixture of Prompts (MoP) strategy to enhance the segmentation robustness. MoP addresses instrument similarity by integrating detailed text descriptions, hence enhancing the classification among instruments. Moreover, we introduce the Hard Instrument Area Reinforcement (HIAR) module that further improves the segmentation in areas where instruments and tissues often overlap. HIAR deepens the model's understanding of challenging regions while reducing confusion between similar instruments.
>
>
> The lack of surgical instrument segmentation datasets indeed poses an overfitting risk.
> A larger dataset could both enhance the model performance and its robustness.
> This however is not a free lunch but comes with much more expensive annotation cost.
> The cross-dataset experiments presented in Tabs. 1 & 2 of the paper demonstrate that our model can bypass this situation with its potential for open-set instrument segmentation.
> For instance, when our model is trained on EndoVis2017, it can actually adeptly handle previously unseen classes, such as suction instrument (SI) in EndoVis2018 by utilizing only their text prompts without retraining.

---

> > ### Comment · Reviewer_9oei · 2023-08-14
> >
> > Hi authors,
> >
> > Thanks for your hard work on the rebuttal.
> >
> > Overall, I am familiar with general segmentation tasks instead of surgical segmentation. It is challenging for me to make the proper justification in this area. Therefore, I have carefully read other reviewers' comments and see their appreciation of this work.
> >
> > Finally, I change my original score to borderline acceptance. I cannot give higher scores due to that I still lack specific knowledge. Hope the SPCs and ACs can see more comments from other reviewers.

---

> > > ### Author Response · Authors · 2023-08-14
> > >
> > > Many thanks for acknowledging our rebuttal and paper, we will continuely improve our paper according to your comments in the revised version.

---

### Official Review · Reviewer_Rt9K · 2023-07-06

**Soundness:** 2 fair
**Presentation:** 2 fair
**Contribution:** 2 fair
**Rating:** 7
**Confidence:** 4

**Summary:**

This paper presents a text prompt-based surgical instrument segmentation method, which is more scalable to handle the diverse targets in endoscopy videos.

**Strengths:**

- originality: this is the first work to use text prompt for surgical instrument segmentation
- quality: the performance of the proposed methods surpassed previous methods
- clarity: the paper is well organized
- significance: the method can handle unseen targets, which is desired in clinical scenarios.

**Weaknesses:**

- EndoVis challenge is organized every year. However, this paper validated the method on old datasets (18-19). Why not use the latest dataset and compare it to the challenge winning solutions? e.g., 21-22
http://opencas.dkfz.de/endovis/challenges/2022/

- Reference formats are not consistent.

**Questions:**

- Does the hard instrument area identified by the model align with the surgeons' perspective?

- Since the best performance in Table 1-2 still has a large room for further improvement. What are the typical failure modes? What's the potential reason for the model to generate such segmentation errors?

**Limitations:**

A more comment task is to segment both instruments and tissues.  It would be great to validate the method in this setting. Here is a public dataset
https://www.kaggle.com/datasets/newslab/cholecseg8k

---

> ### Author Rebuttal · Authors · 2023-08-09
>
> ### [Rt9K-W1] Why not validate the method on the latest EndoVis datasets and compare with the challenge winners?
>
> Please refer to our global response for the justification of choosing Endovis2017 and 2018 datasets.
>
> Our method actually compares with winning solutions, like TernausNet-11 [36] in Tab. 1 of the paper, which was that year's winner. Additionally, we research contemporary papers and compare with SOTA methods like S3Net [5] and MATIS [3], underscoring our approach's superiority.
>
> ### [Rt9K-W2] Reference formats are not consistent.
>
> We will correct it in our revised submission.
>
> ### [Rt9K-Q1] Does the hard instrument area identified by the model align with the surgeons' perspective?
>
> Referring to the Fig. 2 in pdf file in global response section, we visualize the hard instrument areas and observe that these areas predominantly reside at the instrument's clasper and shaft positions (red rectangles).
> For the clasper, due to its deep interaction with the tissue, making the image complex and the segmentation challenging.
> The shaft, on the other hand, presents issues because different instruments often have similar shaft appearances, leading to model misinterpretations.
> After consulting with surgeons, they agreed that the clasper is the hard area to identify, aligning with our findings, but they didn't find the shaft as challenging.
> The difference arises because the instrument clasper, influenced by factors like lighting, can more easily be mistaken for tissue.
> While for surgeons, the classification of the instrument shaft is inferred from the clasper, making the shaft a non-challenging area for them.
> However, for models, grasping the relationship between the clasper and shaft might not be as intuitive, leading to misclassification.
> This observation suggests that future work should focus on modeling the relationship between the clasper and shaft to enhance segmentation performance across different parts.
>
> ### [Rt9K-Q2] Typical failure modes and reasons.
>
> As described in the introduction of the paper, surgical instrument segmentation faces two typical difficulties.
> The first is the continual emergence of new instruments, necessitating frequent model retraining to accommodate these novelties.
> The second pertains to misclassification when segmenting similar instruments and their boundaries.
> Our paper proposes solutions tailored to these challenges: To address the emergence of new surgical instruments, we pioneer the definition of surgical instrument segmentation in a text-promptable format, enabling the open-set segmentation. We also introduce the Mixture of Prompts (MoP) strategy to enhance the segmentation robustness. MoP addresses instrument similarity by integrating detailed text descriptions, hence enhancing the classification among instruments. Moreover, we introduce the Hard Instrument Area Reinforcement (HIAR) module that further improves the segmentation in areas where instruments and tissues often overlap. HIAR deepens the model's understanding of challenging regions while reducing confusion between similar instruments.
>
> Despite these proposed advancements, challenges persist to certain extent (see Fig. 9 & 10 in supplementary material).
> The reason is manifold: the alignment between text and image features is not impeccable, and our current method treats different phrases within prompts uniformly, leading to potential identification challenges for new instruments.
> Additionally, while HIAR alleviates mis-classification issues, it doesn't eradicate them entirely.
> Some ambiguous (hard) areas can be indistinguishable even to human eyes, requiring further exploration for a holistic solution. In the future, we will focus on refining text-image alignment methodologies, leveraging weighting mechanisms to accentuate pivotal distinctions in prompts across instruments, ultimately diminishing segmentation mis-classification, and bolstering instrument edge segmentation through advanced techniques.
>
> ### [Rt9K-L1] It would be great to validate the method on CholecSeg8k.
>
> Thanks for this suggestion. We have done it accordingly, please refer to the CholecSeg8k experiment section in the global response.

---

> > ### Comment · Reviewer_Rt9K · 2023-08-14
> >
> > Thanks for your detailed explanation very much. My major concern has been addressed.
> > One minor question, if the paper is accepted, could you please promise to make the complete training and inference code publicly available by Dec.?

---

> > > ### Author Response · Authors · 2023-08-14
> > >
> > > Thank you very much for your reply. If our paper is accepted, we promise to release the code by December.

---

> > > > ### Comment · Reviewer_Rt9K · 2023-08-18
> > > >
> > > > Thank you for your response.
> > > >
> > > > I will change my decision to accept.

---

> > > > > ### Author Response · Authors · 2023-08-19
> > > > >
> > > > > Thank you very much, we will continue to improve our paper.

---

### Official Review · Reviewer_NUuZ · 2023-07-08

**Soundness:** 3 good
**Presentation:** 3 good
**Contribution:** 3 good
**Rating:** 7
**Confidence:** 4

**Summary:**

The paper introduces a novel approach for surgical instrument segmentation in minimally invasive surgeries. By leveraging text prompts and vision-language models, the proposed method achieves improved segmentation performance. The approach shows promise for practical use in robotic-assisted surgery.


**Strengths:**

The present work contributes with an innovative and effective approach for text promptable surgical instrument segmentation in minimally invasive surgeries.

This paper presents a meticulous study of previous work, which is important in the development of the present work. Also, the technical aspects are clearly explained and have also been evaluated using the correct metrics.

Another strength of this paper is the introduction of a mixture of prompts mechanism. By leveraging multiple text prompts for each surgical instrument, the authors enhance the segmentation performance of their model.

The experimental evaluation of the proposed model on EndoVis2017 and EndoVis2018 datasets demonstrates its superior performance compared with other works and promising generalization capability.

In summary, the work is an interesting application of deep learning in the medical area, and it also has a remarkable novelty.

**Weaknesses:**

Regarding the ablation study, it would be good if the authors could explain why they chose 448x448 as the image size. Aren't some details lost using this size?

It would be good if the authors could give more details about the dataset, i.e. the average duration of each video for example.

It would be very positive to also include more datasets, such as EndoVis2019, EndoVis2020 or EndoVis2021.

**Questions:**

Do you plan to include more datasets such as EndoVis2019, EndoVis2020 or EndoVis2021?

**Limitations:**

limitations are not mentioned, authors should include the limitations in the paper.

---

> ### Author Rebuttal · Authors · 2023-08-09
>
> ### [NUuZ-W1] Why chose $448 \times 448$ as the input image size?
>
> The previous methods use different input image sizes ranging from $224 \times 224$ to original image size (i.e.,$1024 \times 1280$).
> Given our adoption of a ViT-based image encoder, the input size must conform to ViT's patching requirements, i.e. being divisible by 14, 16, and 32 to fit various ViT models.
> As such, we choose $448 \times 448$ as a default setting considering both efficiency and accuracy. Notice that, as with all other methods, our predictions are always reshaped to the original image size for fair evaluation.
>
> ### [NUuZ-W2] More details about the datasets.
>
> Below are detailed descriptions of various datasets.
>
> - EndoVis2017 consists of ten sequences of abdominal porcine procedures, each containing 300 frames sampled at 1 Hz. For training data, the first 225 frames from eight sequences are used, and the remaining 75 frames are kept for testing. Two more sequences with full 300 frames are also reserved for testing.
>
> - EndoVis2018 includes 19 sequences, 15 of them are considered as the training set while the rest 4 as test set. Each sequence is originated from a single porcine training procedure. Redundant frames are manually removed to precisely ensure 300 frames in each sequence.
>
> - EndoVis2019 (Robust MIS) is derived from 30 minimally invasive surgical procedures, including 10 rectal resection, 10 proctocolectomy, and 10 sigmoid resection procedures. A total of 10,040 images are extracted from these procedures. The dataset consists of both training and test cases. Each case contains a 10-second video snippet with 250 endoscopic image frames and a reference annotation for the last frame.
>
> - CholecSeg8K is derived from Cholec80, containing 80 videos of cholecystectomy surgeries performed by 13 surgeons. Each video in CholecSeg8K is recorded at 25 FPS and has annotations for instruments and operation phases. Each video clip contributes 80 image frames, and for each of these frames, the dataset includes raw image data, annotations, and colour masks. In total, the dataset comprises 101 directories with a collection of 8,080 frames.
>
> ### [NUuZ-W3, Q1] Experiments on more datasets.
>
> Thank you for your suggestion. To further validate our approach, we have added experimental results on EndoVis2019 and CholecSeg8k. For detailed information, please refer to our global response section.

---

> > ### Comment · Reviewer_NUuZ · 2023-08-21
> > **Final remarks**
> >
> > Thank you for your comprehensive response to my review of your paper. I appreciate the clarifications and additional details you've provided in response to the concerns I raised. Certainly, your explanations have explained better different aspects of your work.
> >
> > I appreciate your explanation regarding the choice of the input image size (448x448) and your consideration of the requirements imposed by the ViT-based image encoder. The provided information clears up any confusion I had regarding this matter.
> >
> > Thank you for providing detailed descriptions of the datasets used in your experiments. The additional information you've shared regarding EndoVis2017, EndoVis2018, EndoVis2019, and CholecSeg8K is invaluable in understanding the scope and diversity of the data you've employed.
> >
> > I am pleased to see that you've taken our suggestion into consideration and included experimental results on EndoVis2019 and CholecSeg8K. This expansion of your evaluation enhances the robustness and generalizability of your findings, and we believe it will strengthen the paper's contribution.
> >
> > Based on your detailed responses and the additional information you've provided, I am confident that your paper offers a positive contribution to the domain of text-promptable surgical instrument segmentation.

---

> > > ### Author Response · Authors · 2023-08-21
> > >
> > > Thank you very much for your recognition. We will continue to improve our paper and incorporate the information from the rebuttal.

---

> ### Comment · Area_Chair_PXWr · 2023-08-19
>
> This is a friendly reminder from the AC that you need to respond to the rebuttal, since the authors spent quite a lot of time preparing the rebuttal.

---

### Official Review · Reviewer_wnjR · 2023-07-27

**Soundness:** 3 good
**Presentation:** 2 fair
**Contribution:** 3 good
**Rating:** 6
**Confidence:** 4

**Summary:**

This paper introduces a novel idea of utilizing text prompts and vision-language models to make surgical instrument segmentation more flexible and robust to diversity. The proposed method and custom modules achieve strong results on two endoscopic datasets.

**Strengths:**

1. The paper tackles an important problem in surgical instrument segmentation, which aims to enhance robot-assisted surgery systems. The key idea of using text prompts to improve generalization and adaptability to new instruments is novel.
2. The method is technically sound, leveraging recent advances in vision-language models like CLIP. The image and text encoder setup makes sense. The text promptable mask decoder uses attention and convolution schemes nicely for decoding.
3. Several custom modules are proposed to boost segmentation performance: 1) Mixture of prompts leverages multiple prompts effectively. 2) Hard instrument area reinforcement focuses on challenging regions.
4. Comprehensive experiments on two datasets demonstrate superior performance over state-of-the-art methods. The cross-dataset generalization results are promising. The ablation studies validate the efficacy of individual model components like multi-scale feature extraction, mixture of prompts, and hard area reinforcement.

**Weaknesses:**

1. The problem definition and goal can be further sharpened. How does text-based prompting specifically help with increasing instrument variety and subtle inter-class differences? This needs more elaboration upfront.
2. Some architectural details are unclear - like how exactly text features are integrated into the convolutional prompting scheme. More implementation specifics will help reproducibility.
3. The computational complexity and inference speed are not analyzed. This could be important for practical usage.
4. More in-depth experimentation on real-world surgical videos and systems would be preferred to further demonstrate applicability.

**Questions:**

Here are some suggestions for authors to consider:
1. Explain how text prompting helps with adaptation to new instruments and resolving inter-class confusions, with examples.
2. The method sections explain each component logically but can be more crisp and coherent in places. Authors could provide more implementation specifics for text feature integration into convolutional prompting and gating network.
3. Consider reporting computational complexity and inference speeds. How does it compare with prior arts?
4. Consider detecting and segmenting novel instruments not seen during training by using only text prompts.

Typos:
1. line 58: "launguage" should be "language".
2. Mixed use of "visual-textual" and "visual-textural" at multiple places.

Authors should also consider discussing the limitations, such as those listed below, in the paper.

**Limitations:**

1. The method has only been evaluated on two datasets with limited surgical scenarios. Performance on more diverse real-world data is unclear.
2. It relies on high-quality textual prompts, which may not always be available or easy to construct in practice.
3. The requirement of retraining with new text prompts for new instruments reduces adaptability.

---

> ### Author Rebuttal · Authors · 2023-08-09
>
> ### [wnjR-W1, Q1] Explain how text prompting helps with adaptation to new instruments and resolving inter-class confusions, with examples.
>
> For the adaptation to new instruments,
> unlike previous instrument segmentation methods [36, 18, 11, 47, 5, 3] that require model retraining with new data, our text-promptable method instead offers the open-set potential that can directly segment/infer new instruments in images based on their textual descriptions.
> For example, in Tab. 2 of the paper we show that, despite "suction instruments" appearing only in EndoVis2018, our model trained from EndoVis2017 attains a high IoU of 79.77\% (SI in Tab. 2 of the paper). This underscores our approach's adaptability to new instruments.
>
> On the other hand, for resolving inter-class confusion, our text prompting emphasizes unique traits of instrument in descriptions, aiding finer instrument class differentiation.
> For example, while Bipolar forceps and Prograsp forceps are visually similar (see Fig. 1 in pdf file in global response section), they can be distinguished by "elongated tweezer-like design" and "curved scissor-like handles" from their textual descriptions, respectively.
> These textual distinctions can help enhance the visual classification.
>
> ### [wnjR-W2, Q2] More details for text feature integration into convolutional prompting and gating network.
>
> Given the visual and textual feature, $F_I \in \mathbb{R}^{N \times D}$ and $F_T \in \mathbb{R}^{1 \times D}$, respectively, we transform $F_T$ via a fully connected (FC) layer: $\tilde F_{T} = FC(F_{T})$.
> $F_T$ is a vector of dimension $D$, the FC layer reshapes it to the vector $\tilde F_{T}$ of dimension $D \times k \times k + 1$, with "$k \times k$" representing the convolution kernel size and "$+1$" the extra dimension accounting for bias.
> This allows the decomposition of $\tilde F_{T}$ into convolution weights $w \in \mathbb{R}^{1 \times D \times k \times k}$ and bias $b \in \mathbb{R}^{1}$, which are used in subsequent convolutions.
>
> For the gating network, it consists of a 3-layer residual block [13].
> We duplicate $F_T \in \mathbb{R}^{1 \times D}$ to $F_T^p \in \mathbb{R}^{N \times D}$ to match $F_I$'s dimension and concatenate them before inputting to $\mathcal G$. The output of $\mathcal G$ are three weight maps corresponding to the three score maps in $\mathcal S$.
> These weights are normalized using softmax operations along the prompt dimension (they are normalized pixel-wisely across weight maps).
> We calculate the weighted sum of scores maps in $\mathcal S$ to derive the final score map.
>
> In the revised version, we will refine our paper and release the code.
>
> ### [wnjR-W3, Q3] Analysis of computational complexity and inference speed.
>
> We assess the computational complexity and inference speed by evaluating floating point operations per second (FLOPs) and frames per second (FPS) respectively, using a single A100 GPU. We run the experiment on EndoVis2017 by resizing the input image to the default input sizes corresponding to different methods (e.g. $800 \times 800$ for ISINet, $224 \times 224$ for MATIS, $416 \times 416$ for CRIS, $448 \times 448$ for CLIPSeg and Ours).
> From the table below, it's evident that our model's computational complexity (FLOPs) and inference speed (FPS) align with other adapted text-promptable approaches (i.e., CRIS and CLIPSeg), achieving real-time performance suitable for clinical applications.
> On the other hand, compared to conventional segmentation methods (i.e., ISINet and MATIS), ours appears to be clearly more efficient than ISINet; while it is marginally slower than MATIS [3], likely due to MATIS's small input size.
>
>  Method       | FLOPs (G) | FPS
> --------------|-----------|-----
>  ISINet [11]  | 264       | 19
>  MATIS [3]    | 66        | 27
>  CRIS [39]    | 196       | 19
>  CLIPSeg [24] | 127       | 23
>  Ours         | 125       | 22
>
> ### [wnjR-W4, L1] Experiments on more diverse real-world data.
>
> Thank you for your suggestion.
> The data in EndoVis2017 and 2018 datasets are indeed from real-world surgical videos.
> To further validate our approach, we have added experimental results on EndoVis2019 and CholecSeg8k, which both consist of data from real-world surgical videos.
> For detailed information, please refer to our global response section.
>
> ### [wnjR-Q4] Use textural prompts to segment unseen instruments.
>
> In our study, the cross-dataset experiments between EndoVis2017 and EndoVis2018 (Tabs. 1 & 2 of the paper) indeed underscore the potency of our method for segmenting unseen instruments.
> For instance, when our model is trained on EndoVis2017, it can actually adeptly handle previously unseen classes, such as suction instrument (SI) in EndoVis2018 by utilizing on only their textural prompts without retraining.
>
> ### [wnjR-L2] High-quality textual prompts potentially limit practical use.
>
> Firstly, we have designed well-crafted question templates to guide LLMs like GPT to automatically derive high-quality textural prompts (see Section 6.1 in our supplementary material).
> Given these predefined question templates, adapting to new instruments becomes straightforward.
> Secondly, our method with simple prompts (not using LLMs) still performs very well (see Tab. 5 in the paper), outperforming SOTA with large margins.
> Thirdly, once our model is trained, it is equipped with the open-set potential that can segment new instruments without retraining/finetuning (see cross dataset experiment in Sec 4.3).
>
> ### [wnjR-L3] Retraining with new prompts for different instruments limits adaptability.
>
> As highlighted in [wnjR-W1, Q1] and [wnjR-L2], our text-promptable approach can adapt to new instruments without retraining/finetuning the model.
>
> ### [wnjR-Misc] Typos.
> Thanks! We will rectify them in the final version of the paper.

---

> > ### Comment · Reviewer_wnjR · 2023-08-17
> >
> > Thank you for the detailed rebuttal. I appreciate the additional experimental results and analysis. Please try to incorporate the details mentioned in the rebuttal into the revised version. Since more datasets are involved, I think it will be beneficial to also include more cross-dataset results to better demonstrate the robustness of the proposed method for segmenting unseen instruments. I presume it is relatively easy to complete as no retraining is needed.
> >
> > In general, the authors' rebuttal resolved most of my concerns. I'm raising my rating to weak accept.

---

> > > ### Author Response · Authors · 2023-08-17
> > >
> > > Thank you very much for recognizing the value of our rebuttal.
> > > We will include the additional experimental results from the rebuttal phase in our revised paper.
> > >
> > > We appreciate your suggestion, and we also plan to incorporate more cross-dataset validation results in the revised paper.
> > > We conducted a quick cross-dataset experiment using a model trained on EndoVis2017 and tested it on the EndoVis2019 dataset.
> > > Our method achieved DSC=0.90 and NSD=0.90, which surpasses the competition's previous best result of DSC=0.89 and NSD=0.89.
> > > This further underscores the exceptional performance of our approach.
> > > We will provide more cross-dataset experimental results in our subsequent revised paper.

---

### Author Rebuttal · Authors · 2023-08-09

We thank the constructive comments from all reviewers.
As reviewers say, our key idea of using textural prompts to perform surgical instrument segmentation is novel (wnjR, NUuZ) and interesting (NUuZ). Our paper is overall well organized (Rt9K) and easy to understand (9oei).
Below we first answer the common question raised by several reviewers on our experimental datasets; next, we address the concerns from each reviewer separately.

## Experiments on more datasets

Several reviewers point out the need to validate our method on more datasets.
We deeply value your comments.
First, we would like to emphasize that it is a common practice [36, 18, 11, 47, 5, 3] to evaluate the surgical instrument segmentation performance on the two established datasets, EndoVis 2017 [1] and 2018 [2].
We followed this practice and should be not disadvantaged.
Next, we delineate the consideration for selecting additional datasets, and then report our results on these datasets.

### Dataset selection
Although the EndoVis Challenge is an annual event, datasets specific to surgical instrument segmentation aren't released every year.
Specifically, the EndoVis2019's Robust-MIS dataset merely differentiates between tissues and instruments, not aligning with our study's focus on segmenting different instrument types.
Moreover, the instance segmentation in EndoVis2019 also does not align with our paper's problem.
The challenges in EndoVis2020 and 2021 do not address instrument segmentation either.
Regarding the datasets for EndoVis2022 and 2023 challenges, namely SAR-RARP50 and SIMS, their usage in publications is restricted until the release of their respective competition reports.
Given the SAR-RARP50 report's pending release and the ongoing SIMS competition, these datasets are not incorporated into our research.

To further substantiate our approach, we choose to conduct experiments on the instrument binary segmentation task on the EndoVis2019 dataset [R1], as well as instrument and tissue segmentation task on the CholecSeg8k dataset [R2].

### Comparison to state of the art

1. **EndoVis2019**: Consistent with the competition's evaluation protocol [R1], we use the Dice Similarity Coefficient (DSC) and Normalized Surface Dice (NSD) to assess the segmentation performance.
As the following table shows, our method (input size 448) notably surpasses the competition's top performers, with +3% increase in DSC and +2% enhancement in NSD, which demonstrates the superiority of our method. It's worth noting that our approach is designed for multi-class segmentation while is tested for binary class segmentation. Despite this, the performance improvement by ours over SOTA underscores its efficacy.

 Method         | DSC  | NSD
----------------|------|------
 haoyun [R1]    | 0.89 | 0.89
 CASIA-SRL [R1] | 0.78 | 0.89
 Ours (448)           | 0.92 | 0.91

2. **CholecSeg8k**: Following the protocols from SP-TCN [R4], we split the dataset into training and testing sets (videos 12, 20, 48 and 55 for testing and others for training) and utilize the mean Intersection over Union (mean IoU) as the evaluation metric.
In the following table, the result of our method is 71.03% mean IoU.
It's evident that our method surpasses the current SOTA (SP-TCN) by 1.65% in mean IoU, even though SP-TCN leverages temporal information from videos to boost the performance, while our method solely relies on individual image data.
It's worth noting that for the CholecSeg8k, we use the same prompt generation method described in our paper to obtain prompts for both tissues and instruments. The result demonstrates that prompts for tissues are appropriately generated following our method, further attesting to the generalizability of our method.

 Method                  | mean IoU
-------------------------|----------
 Swin base [R3]          | 0.6842
 Swin base + SP-TCN [R4] | 0.6938
 Ours (448)                    | 0.7103

## References
[R1] Ross, Tobias, et al. "Robust medical instrument segmentation challenge 2019." arXiv preprint arXiv:2003.10299 (2020).

[R2] Hong, W-Y., et al. "Cholecseg8k: a semantic segmentation dataset for laparoscopic cholecystectomy based on cholec80." arXiv preprint arXiv:2012.12453 (2020).

[R3] Liu, Ze, et al. "Swin transformer: Hierarchical vision transformer using shifted windows." Proceedings of the IEEE/CVF International Conference on Computer Vision. 2021.

[R4] Grammatikopoulou, Maria, et al. "A spatio-temporal network for video semantic segmentation in surgical videos." International Journal of Computer Assisted Radiology and Surgery (2023): 1-8.

---

### Decision · Program_Chairs · 2023-09-21

**Decision:**

Accept (poster)

**Comment:**

After rebuttal, all reviewers land on the positive side. The authors well addressed the main concerns raised by reviewers, e.g., the lack of experiments on more challenging datasets. Thus, the AC recommends acceptance. The authors should include the discussion and new results in the rebuttal into the final version. Besides, the authors should release the code as they promised in the rebuttal.